# Learning Continuous Normalizing Flows For Faster Convergence To Target Distribution via Ascent Regularizations

**Shuangshuang Chen**[1,2]  **Sihao Ding**[2]  **Yiannis Karayiannidis**[3]  **Mårten Björkman**[1]

[1] School of Electrical Engineering and Computer Science, Royal Institute of Technology
[2] AI Lab, Volvo Car Corporation
[3] Department of Automatic Control, Lund University
{shuche, celle}@kth.se, sihao.ding@volvocars.com, yiannis@control.lth.se

## Abstract

Normalizing flows (NFs) have been shown to be advantageous in modeling complex distributions and improving sampling efficiency for unbiased sampling. In this work, we propose a new class of continuous NFs, ascent continuous normalizing flows (ACNFs), that makes a base distribution converge faster to a target distribution. As solving such a flow is non-trivial and barely possible, we propose a practical implementation to learn flexibly parametric ACNFs via ascent regularization and apply it in two learning cases: maximum likelihood learning for density estimation and minimizing reverse KL divergence for unbiased sampling and variational inference. The learned ACNFs demonstrate faster convergence towards the target distributions, therefore, achieving better density estimations, unbiased sampling and variational approximation at lower computational costs. Furthermore, the flows show to stabilize themselves to mitigate performance deterioration and are less sensitive to the choice of training flow length $T$.

## 1 Introduction

Normalizing flows (NFs) provide a flexible way to define an expressive but tractable distribution which only requires a base distribution and a chain of bijective transformations (Papamakarios et al., 2021). Neural ODE (Chen et al., 2018) extends discrete normalizing flows (Dinh et al., 2014; 2016; Papamakarios et al., 2017; Ho et al., 2019) to a new continuous-time analogue by defining the transformation via a differential equation, substantially expanding model flexibility in comparison to the discrete alternatives. (Grathwohl et al., 2018; Chen and Duvenaud, 2019) propose a computationally cheaper way to estimate the trace of Jacobian to accelerate training, while other methods focus on increasing flow expressiveness by e.g. augmenting with additional states (Dupont et al., 2019; Massaroli et al., 2020), or adding stochastic layers between discrete NFs to alleviate the topological constraint (Wu et al., 2020). Recent diffusion models like (Hodgkinson et al., 2020; Ho et al., 2020; Song et al., 2020; Zhang and Chen, 2021) extend the scope of continuous normalizing flows (CNFs) with stochastic differential equations (SDEs). Although these diffusion models significantly improve the quality of the generated images, the introduced diffusion comes with costs: some models no longer allow for tractable density estimation; or the practical implementations of these models rely on a long chain of discretizations, thus needing relatively more computations than tractable CNF methods, which can be critical for some use cases such as online inference.

(Finlay et al., 2020; Onken et al., 2021; Yang and Karniadakis, 2020) introduce several regularizations to learn simpler dynamics using optimal transport theory, which decrease the number of discretization steps in integration and thus reduce training time. (Kelly et al., 2020) extends the $L_2$ transport cost to regularize any arbitrary order of dynamics. Although these regularizations are beneficial for decreasing the computational costs of simulating flows, they do not improve the slow convergence of density to the target distributions like trained vanilla CNF models shown in Figure 1. To accelerate the flow convergence, STEER (Ghosh et al., 2020) and TO-FLOW (Du et al., 2022) propose to optimize flow length $T$ in two different approaches: STEER randomly samples the length during training while TO-FLOW establishes a subproblem for $T$ during training. To understand the effectiveness of

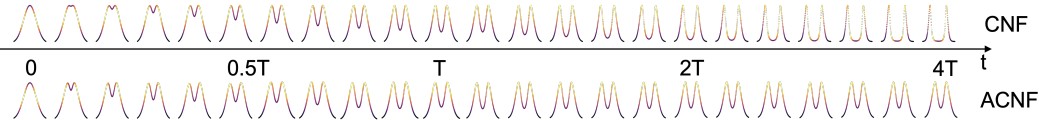

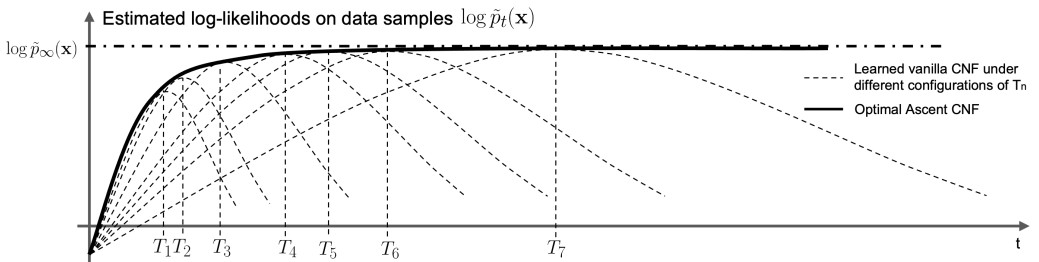

Figure 1: Distribution transformations of two learned flows for 1d Gaussian mixture from a Gaussian distribution at $t \in [0, 4T]$. Although the two flows reach similar densities at $T$, the density of ACNF converges faster to the target distribution before $T$ and diverges slower after $T$ than that of CNF. Color indicates the density of true Gaussian mixture.

Figure 2: The log-likelihood estimates of trained vanilla CNF models with various flow length $T_n$ and the steepest ACNF with dynamics defined in eq.(6) at different $t$ on 2-moon distribution. All vanilla CNF models reach their maximum around $T_n$ and deteriorate rapidly afterwards while the log-likelihood estimate of ACNF elevates rapidly at initial and increases monotonically.

these methods, we train multiple Neural ODE models with different flow length $T_n$ for a 2-moon distribution and examine these flows by the estimated log-likelihoods in Figure 2. Although sampling or optimizing $T$ dynamically performs a model selection during training and leads models to reach higher estimates at shorter flows, it cannot prevent the divergence after $T_n$. Furthermore, shorter flows are more limited in expressiveness for higher maximum likelihoods and sensitive to flow length.

In this work, we present a new family of CNFs, ascent continuous normalizing flows (ACNFs), to address the aforementioned problems. ACNF concerns a flow that transforms a base distribution monotonically to a target distribution, and the dynamics is imposed to follow the steepest ACNF. However, solving such a steepest flow is non-trivial and barely possible. We propose a practical implementation to learn parametric ACNFs via ascent regularization. Learned ACNFs exhibit three main beneficial behaviors: 1) faster convergence to target distribution with less computation; 2) self-stabilization to mitigate flow deterioration; and 3) insensitivity to flow training length $T$. We demonstrate these behaviors in three use cases: modeling data distributions; learning annealed samplers for unbiased sampling; and learning a tractable but more flexible variational approximation.

## 2  CONTINUOUS NORMALIZING FLOWS

Considering a time-$t$ transformation $\mathbf{z}(t) = \boldsymbol{\Phi}_t(\mathbf{x})$ on the initial value $\mathbf{x}$, i.e. $\mathbf{z}(0) = \mathbf{x}$, the change of variable theorem reveals the relation between the transformed distribution $p_t(\mathbf{z}(t))$ and $p(\mathbf{x})$:

$$p_t(\mathbf{z}(t)) = \left| \det \left( \mathcal{J}_{\boldsymbol{\Phi}_t}^{-1}(\mathbf{x}) \right) \right| p(\mathbf{x}), \tag{1}$$

where $\mathcal{J}_{\boldsymbol{\Phi}_t}$ is the Jacobian matrix of $\boldsymbol{\Phi}_t$. As $\boldsymbol{\Phi}_t$ normalizes $\mathbf{x}$ towards some base distribution, $p_t(\mathbf{z}(t))$ is referred to as the normalized distribution at time $t$, starting from the data distribution $p(\mathbf{x})$.

Continuous normalizing flow is the infinitesimal limit of the chain of discrete flows and the infinitesimal transformation is specified by an ordinary differential equation (ODE):

$$\frac{d\mathbf{z}(t)}{dt} = \frac{d\boldsymbol{\Phi}_t(\mathbf{x})}{dt} = \mathbf{f}(\mathbf{z}(t), t). \tag{2}$$

The instantaneous change of variable theorem (Chen et al., 2018, theorem 1) shows the infinitesimal changes of $\log p_t(\mathbf{z}(t))$ is:

$$\frac{d \log p_t(\mathbf{z}(t))}{dt} = -\boldsymbol{\nabla} \cdot \mathbf{f}(\mathbf{z}(t), t). \tag{3}$$

Figure 3: Upper: transformations on variables and densities on normalization and sampling directions. Lower left: data samples (orange) and the grid of states (blue) transformations along normalization direction. Lower right: density estimation $\tilde{p}_t$ along sampling direction.

Thus, the log-normalized distribution $\log p_t(\mathbf{z}(t))$ can be obtained by integrating eq.(3) backwards with a common approximation to the base distribution $\mu$, i.e. $p_T \approx \mu$:

$$\log p_t(\mathbf{z}(t)) = \log p_T(\mathbf{z}(T)) - \int_t^T \boldsymbol{\nabla} \cdot \mathbf{f}(\mathbf{z}(\tau), \tau) \mathrm{d}\tau \approx \log \mu(\mathbf{z}(T)) - \int_t^T \boldsymbol{\nabla} \cdot \mathbf{f}(\mathbf{z}(\tau), \tau) \mathrm{d}\tau,$$

where $\mathbf{z}(t) = \mathbf{x} + \int_0^t \mathbf{f}(\mathbf{z}(\tau), \tau) \mathrm{d}\tau$. The accuracy of $\log p_0(\mathbf{x})$, obtained by the right hand side, depends on the approximation error of $p_T$ to $\mu$ and the error can vary at different $\mathbf{z}(T)$. To avoid the problems in analysis and investigate how flow length affects on modeled distribution, we introduce $\tilde{p}_t(\mathbf{x})$, estimating density of a $t$-length flow $\boldsymbol{\Phi}_t$, which is shown via the change of variable theorem:

$$\tilde{p}_t(\mathbf{x}) = \left|\det\left(\mathcal{J}_{\boldsymbol{\Phi}_t}(\mathbf{x})\right)\right| \mu(\boldsymbol{\Phi}_t(\mathbf{x})). \tag{4}$$

As indicated by eq.(4) and Figure 3, $\tilde{p}_t$ initiates at the base distribution, i.e. $\tilde{p}_0(\mathbf{x}) = \mu(\mathbf{x})$. Combining eq.(1) and eq.(4), the estimated density $\tilde{p}_t$ relates to normalized distribution $p_t(\mathbf{z}(t))$ as:

$$\frac{\tilde{p}_t(\mathbf{x})}{p(\mathbf{x})} = \frac{\mu(\boldsymbol{\Phi}_t(\mathbf{x}))}{p_t(\boldsymbol{\Phi}_t(\mathbf{x}))} = \frac{\mu(\mathbf{z}(t))}{p_t(\mathbf{z}(t))}.$$

It shows that as $p_t \to \mu$, $\tilde{p}_t(\mathbf{x}) \to p(\mathbf{x})$. When there exists a flow, of which the normalized density is equal to the base distribution, i.e. $p_T = \mu$, then the estimated likelihood becomes exact to the data distribution, i.e. $\tilde{p}_T(\mathbf{x}) = p(\mathbf{x})$. Like the instantaneous change of variable theorem in eq.(3), we derive the infinitesimal change of time-$t$ estimated log-likelihood:

**Proposition 1** (Instantaneous Change of Log-likelihood Estimate). *Let $\mathbf{z}(t)$ be a finite continuous random variable at time $t$ as the solution of a differential equation $\frac{\mathrm{d}\mathbf{z}(t)}{\mathrm{d}t} = \mathbf{f}(\mathbf{z}(t), t)$ with initial value $\mathbf{z}(0) = \mathbf{x}$. Assuming that $\tilde{p}_0 = \mu$ at $t = 0$ and $\mathbf{f}$ is uniformly Lipschitz continuous in $\mathbf{z}$ and $t$, then the change in estimated log-likelihood $\log \tilde{p}_t(\mathbf{x})$ at $t$ follows a differential equation:*

$$\frac{\mathrm{d} \log \tilde{p}_t(\mathbf{x})}{\mathrm{d}t} = \boldsymbol{\nabla} \cdot \mathbf{f}(\mathbf{z}(t), t) + \nabla \log \mu(\mathbf{z}(t)) \cdot \mathbf{f}(\mathbf{z}(t), t). \tag{5}$$

*Proof.* See Appendix A.1 for detailed derivation and its relation to eq.(3). Unlike the integral for $\log p_t(\mathbf{z}(t))$ that relies on the approximation and requires to solve the whole trajectory $\mathbf{z}(\tau), \tau \in [0, T]$, $\log \tilde{p}_t(\mathbf{x})$ can be evaluated exactly simultaneously with $\mathbf{z}(t)$ for any or/and different $t$:

$$\log \tilde{p}_t(\mathbf{x}) = \log \mu(\mathbf{x}) + \int_0^t \left(\boldsymbol{\nabla} \cdot \mathbf{f}(\mathbf{z}(\tau), \tau) + \nabla \log \mu(\mathbf{z}(\tau)) \cdot \mathbf{f}(\mathbf{z}(\tau), \tau)\right) \mathrm{d}\tau.$$

## 3 ASCENT CONTINUOUS NORMALIZING FLOWS

By using KL divergence as distance measure of distributions, we have the following duality:

$$\mathrm{KL}(p(\mathbf{x})||\tilde{p}_t(\mathbf{x})) = \mathrm{const} - \int p(\mathbf{x}) \log \tilde{p}_t(\mathbf{x}) \mathrm{d}\mathbf{x} = \mathrm{KL}(p_t(\mathbf{z}(t))||\mu(\mathbf{z}(t))),$$

that maximum likelihood learning of $\tilde{p}_T(\mathbf{x})$ for data samples from $p(\mathbf{x})$ is equivalent to minimizing 1) the forward KL divergence between $p(\mathbf{x})$ and $\tilde{p}_t(\mathbf{x})$ as the first equality; 2) the reverse KL divergence in normalization direction as the second equality. We can measure the rates of KL divergences or the expected log-likelihood by their time derivative, and define ascent continuous normalizing flows (ACNFs) that monotonically decrease KL divergence or increase the expected log-likelihood , i.e.

$$\frac{\partial}{\partial t} \int p(\mathbf{x}) \log \tilde{p}_t(\mathbf{x}) \mathrm{d}\mathbf{x} \geq 0; \text{ or } \frac{\partial}{\partial t} \mathrm{KL}(p_t(\mathbf{z}(t))||\mu(\mathbf{z}(t))) \leq 0.$$

By applying total variation, we can find the dynamics for the steepest descent of reverse KL divergence or the steepest ascent of the expected log-likelihood:

**Theorem 1** (Dynamics for Steepest Ascent Continuous Normalizing Flows). *Let $\mathbf{z}(t)$ be a finite continuous random variable and the solution of a differential equation $\frac{d\mathbf{z}(t)}{dt} = \mathbf{f}(\mathbf{z}(t), t)$ with initial value $\mathbf{z}(0) = \mathbf{x}$. Its probability $p_t(\mathbf{z}(t))$ subjects to the continuity equation $\partial_t p_t + \boldsymbol{\nabla} \cdot (p_t \mathbf{f}) = 0$. The dynamics of the steepest flow for decreasing $KL(p_t(\mathbf{z}(t))\|\mu(\mathbf{z}(t)))$ is*

$$\mathbf{f}^*(\mathbf{z}(t), t) = \nabla \log \mu(\mathbf{z}(t)) - \frac{\nabla p_t(\mathbf{z}(t))}{p_t(\mathbf{z}(t))} = \nabla \log \mu(\mathbf{z}(t)) - \nabla \log p_t(\mathbf{z}(t)). \tag{6}$$

*Proof.* See Appendix A.2 for detailed derivation. The steepest dynamics is the difference between two gradients: $\nabla \log \mu$ and $\nabla \log p_t$ w.r.t. state $\mathbf{z}(t)$. There are a few important implications of eq.(6): 1) the dynamics is time-variant as $p_t$ evolves along the flow of $\mathbf{z}(t)$; 2) the dynamics at time $t$ only depends on the current state $\mathbf{z}(t)$, thus no history is needed; 3) the flow is initiated at the difference between $\nabla \log \mu(\mathbf{x})$ and $\nabla \log p(\mathbf{x})$, gradually slows down and eventually stops when $p_t$ converges to $\mu$. The convergence rate of the steepest flow can also be proven as the negative Fisher divergence, $\partial \mathrm{KL}(p_t \| \mu)/\partial t = -\mathbb{F}(p_t \| \mu) = -\mathbb{E}_{p_t} \|\nabla \log \mu(\mathbf{z}) - \log p_t(\mathbf{z})\|_2^2$, therefore this optimal deterministic CNF is related to (overdamped) Langevin diffusion, see Appendix A.3 for the derivation of convergence rate and detailed discussion of their relation.

This optimal flow also can be considered as a special instance of Wasserstein gradient flow (Ambrosio et al., 2005) with KL divergence as the energy functional. Previous works (Finlay et al., 2020; Yang and Karniadakis, 2020; Onken et al., 2021) apply the optimal transport theory to regularize flow dynamics in Euclidean space, while Wasserstein gradient flow or eq.(6 instead regularizes flow in probability measure space. We refer readers to (Ambrosio et al., 2005) for accessible introduction. In some special cases, the flow can be solved by introducing an auxiliary potential, $V(\mathbf{z}, t) = p_t(\mathbf{z})/\mu(\mathbf{z})$, which has a partial differential equation (PDE):

$$\frac{\partial V(\mathbf{z}, t)}{\partial t} = \Delta V(\mathbf{z}, t) + 2\nabla \log \mu(\mathbf{z}) \cdot \nabla V(\mathbf{z}, t) + \nabla \log V(\mathbf{z}, t) \cdot \nabla V(\mathbf{z}, t), \tag{7}$$

with the initial condition $V(\mathbf{z}(0), 0) = \frac{p_0(\mathbf{z}(0))}{\mu(\mathbf{z}(0))} = \frac{p(\mathbf{x})}{\mu(\mathbf{x})}$. See Appendix A.4 for its derivation. Solving this PDE for $p_t(\mathbf{z}(t))$ is non-trivial as the closed form solution is typically unknown. JKO integration is practically used in literature (Mokrov et al., 2021; Fan et al., 2021) for the solution, which approximates the dynamics of density $p_t$ by its time discretization. However, it requires to know the initial condition while $p(\mathbf{x})$ is generally unknown and needs to be modeled for data. (Tabak and Vanden-Eijnden, 2010) proposes to approximate $p(\mathbf{x})$ by the spatial discretization of samples, which hardly can be scaled up even for intermediate dimensions.

To tackle these difficulties and accelerate unregulated flows for faster convergence, we propose ascent regularization to learn parametric ACNFs, as inspired by previous works (Yang and Karniadakis, 2020; Onken et al., 2021; Finlay et al., 2020; Kelly et al., 2020; Ghosh et al., 2020) that enforce flows with certain behaviors via regularization in training. Ascent regularization penalizes the difference between the parametric dynamics and the steepest dynamics by $\|\mathbf{f}_{\boldsymbol{\theta}} - \mathbf{f}^*\|_2^2$, which needs to evaluate score function $\nabla \log p_t(\mathbf{z}(t))$. Therefore, we propose the instantaneous change of the score function:

**Theorem 2** (Instantaneous Change of Score Function). *Let $\mathbf{z}(t)$ be a finite continuous random variable with probability density $p_t(\mathbf{z}(t))$ at time $t$. Let $\frac{d\mathbf{z}(t)}{dt} = \mathbf{f}(\mathbf{z}(t), t)$ be a differential equation describing a continuous-in-time transformation of $\mathbf{z}(t)$. Assuming that $\mathbf{f}$ is uniformly Lipschitz continuous in $\mathbf{z}$ and $t$, the infinitesimal change in the gradient of log-density at $t$ is*

$$\frac{d\nabla \log p_t(\mathbf{z}(t))}{dt} = -\nabla \log p_t(\mathbf{z}(t)) \frac{\partial \mathbf{f}(\mathbf{z}(t), t)}{\partial \mathbf{z}(t)} - \nabla (\boldsymbol{\nabla} \cdot \mathbf{f}(\mathbf{z}(t), t)). \tag{8}$$

*Proof.* See Appendix A.5 for detailed derivation. $\nabla \log p(\mathbf{z}(t), t)$ follows a linear matrix differential equation, where the linear coefficient is the Jacobian and the bias term is the gradient of divergence. To be noted, an alternative proof can be found in concurrent work (Lu et al., 2022, theorem D.1).

We discuss the training of ACNFs in two different learning cases: maximum likelihood learning for data modeling and density estimation in Section 4; minimizing reverse KL divergence for learning annealed samplers for unbiased sampling in Section 5 .

---

**Algorithm 1** Maximum likelihood learning of ACNF with ascent regularization

**Require:** Data samples $\mathbf{X} = \{\mathbf{x}^j\}_{j=1,\ldots,M}$, parameteric dynamics of flow $\mathbf{f}_{\boldsymbol{\theta}}$, length of flow $T$, ascent regularization coefficient $\lambda$, mini-batch size $N$, base distribution $\mu$
  **Initialize** $\boldsymbol{\theta}$
  **while** $\boldsymbol{\theta}$ is not converged **do**
    Sample a mini-batch of $N$ data $\mathbf{x}^i \sim \mathbf{X}$
    Integrate augmented states
$[\mathbf{z}^i(t), \log \tilde{p}_t(\mathbf{x}^i)]$ forward with initial value $[\mathbf{x}^i, \log \mu(\mathbf{x}^i)]$ from 0 to $T$
    Integrate augmented states
$[\mathbf{z}^i(t), \nabla \log p_t(\mathbf{z}^i(t))]$ backwards with initial value $[\mathbf{z}^i(T), \nabla \log \mu(\mathbf{z}^i(T))]$ from $T$ to 0
    Compute loss function $\mathcal{L}$ in eq.(9) and $\nabla_{\boldsymbol{\theta}} \mathcal{L}$ by adjoint sensitivity method
    Update $\boldsymbol{\theta}$ by gradient descent algorithm
  **end while**

**Algorithm 2** Training ACNF as annealed sampler for unbiased sampling with ascent regularization

**Require:** target distribution $\pi = \gamma/Z$, parameteric dynamics of flow $\mathbf{f}_{\boldsymbol{\theta}}$, length of flow $T$, number of samples $N$, ascent regularization coefficient $\lambda$, base distribution $\mu$
  **Initialize** $\boldsymbol{\theta}$
  **while** $\boldsymbol{\theta}$ is not converged **do**
    Sample $\mathbf{z}^i(0) \sim p_0 = \mu$
    Evaluate $\log \mu(\mathbf{z}^i(0))$ and $\nabla \log \mu(\mathbf{z}^i(0))$
    Integrate augmented states
$[\mathbf{z}^i(t), \log p_t(\mathbf{z}^i(t)), \nabla \log p_t(\mathbf{z}^i(t))]$ with initial value $[\mathbf{z}_0^i, \log \mu(\mathbf{z}_0^i), \nabla \log \mu(\mathbf{z}_0^i)]$ from 0 to $T$
    Evaluate
$\log w(\mathbf{z}^i(T)) = \log \gamma(\mathbf{z}^i(T)) - \log p_T(\mathbf{z}^i(T))$
    Compute loss function $\mathcal{L}$ in eq.(10) and $\nabla_{\boldsymbol{\theta}} \mathcal{L}$ by adjoint sensitivity method
    Update $\boldsymbol{\theta}$ by gradient descent algorithm
  **end while**

---

## 4 MAXIMUM LIKELIHOOD LEARNING OF ACNF FOR DENSITY ESTIMATION VIA ASCENT REGULARIZATION

For maximizing likelihood learning of $\tilde{p}_T$ to fit data, the total objective with ascent regularization is:

$$\min_{\mathbf{f}} \mathcal{L} = \frac{1}{N} \sum_{i=1}^{N} \left( -\log \tilde{p}_T(\mathbf{x}^i; \boldsymbol{\theta}) + \lambda \int_0^T \| \left( \nabla \log p_t(\mathbf{z}^i(t); \boldsymbol{\theta}) - \nabla \log \mu(\mathbf{z}^i(t)) \right) + \mathbf{f}(\mathbf{z}^i(t), t; \boldsymbol{\theta}) \|_2^2 \mathrm{d}t \right),$$

(9)

where $\lambda$ is the ascent regularization coefficient to control the trade-off between maximizing likelihood and regularization on the ascent behavior of the learned dynamics. When $\lambda = 0$, ACNF degrades to vanilla CNF. The first term in eq.(9) is obtained by integrating eq.(5) over $[0, T]$, simultaneously with $\mathbf{z}(t)$, while the ascent regularization can be integrated backwards with augmented initial $[\mathbf{z}(T), \nabla \log p_T(\mathbf{z}(T))]$, with $\nabla \log p_T(\mathbf{z}(T)) \approx \nabla \log \mu(\mathbf{z}(T))$. We summarize the pseudo-code for maximum likelihood learning of ACNFs in Algorithm 1. We show the interpretation of ascent regularization as score matching in Section A.6 in Appendix, thus Algorithm 1 can be implemented in more efficient ways like (Lu et al., 2022; Song et al., 2021) for some cases.

## 5 LEARNING ACNF AS ANNEALED SAMPLER FOR UNBIASED SAMPLING

Except modeling data samples and performing density estimation, NF as a sampler shows to be more sample efficient in Annealed Importance Sampling (AIS) (Neal, 2001) when comparing to classic MCMC methods (Arbel et al., 2021). A typical AIS and its extension use a sequence of annealed targets $\{\pi_k\}_{k=0:K}$ that bridges an easy-to-sample and tractable distribution $\pi_0 = \mu$ to the target $\pi_K := \pi = \gamma(\cdot)/Z$ that is known up to the normalization constant. SNF (Wu et al., 2020) and AFT (Arbel et al., 2021) propose to fit $K$ discrete NFs and each NF approximates the transport map between $\pi_{k-1}$ and $\pi_k$. However, the rate of sampling convergence is dependent on the pre-defined annealed targets. Besides, a larger $K$ annealing step is needed to decrease the variance of the estimator, which comes at an additional computational cost (Doucet et al., 2022).

As ACNF can also define the flow from a base distribution to a target distribution, it can learn a continuous flow of the annealed target instead of the pre-defined discrete one, and later generate samples. Different to (Grosse et al., 2013), the annealed target by ACNF does not require a specific form of distribution. As ACNF enforces faster convergence to the target distribution, ACNF sampler potentially generates better samples than CNF or linear annealed scheduling especially at limited steps $K$, thus the estimate, e.g. on logarithm of normalization constant $\log Z$, is more accurate.

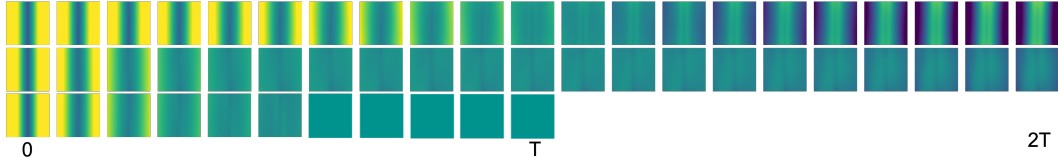

Figure 4: Comparison on log potential field along the flow by trained vanilla CNF and ACNF with $\lambda = 1$ and the numerical PDE solutions of eq.(7) for 2-modal Gaussian mixture at $t \in [0, 2T]$ . Color indicates the value of field: turquoise is 0 and the lighter the color is the larger the value is.

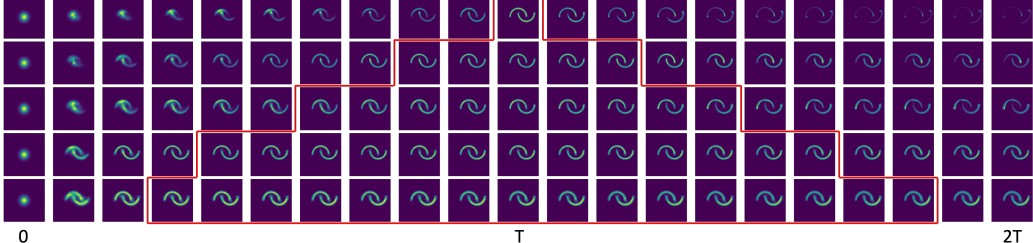

Figure 5: Comparison on density evaluation of trained vanilla CNF and ACNFs with $\lambda = 0.0001, 0.0005, 0.001, 0.005$ on 2-moon distribution along integral $t \in [0, 2T]$.

Different to maximum likelihood in Section 4, training ACNF for annealed sampling is to minimize the reverse KL divergence, $\mathrm{KL}(p_T(\mathbf{z}(T)) \| \pi(\mathbf{z}(T)))$. It can be evaluated up to a constant by the logarithm of importance weights of samples and $\log w(\mathbf{z}^i(T)) = \log \gamma(\mathbf{z}^i(T)) - \log p_T(\mathbf{z}^i(T))$. With ascent regularization like previous section, the total objective becomes:

$$\min_{\mathbf{f}} \mathcal{L} = \frac{1}{N} \sum_{i=1}^{N} \left( -\log w(\mathbf{z}^i(T); \boldsymbol{\theta}) + \lambda \int_0^T \| \left( \nabla \log p_t(\mathbf{z}^i(t); \boldsymbol{\theta}) - \nabla \log \mu(\mathbf{z}^i(t)) \right) + \mathbf{f}(\mathbf{z}^i(t), t; \boldsymbol{\theta}) \|_2^2 \mathrm{d}t \right),$$

(10)

where $\mathbf{f}(\mathbf{z}(t), t; \boldsymbol{\theta})$ is the annealed generation dynamics. Unlike the previous section, as the sampler initiates by the base distribution, $\log p_t(\mathbf{z}^i(t))$ and $\nabla \log p_t(\mathbf{z}^i)$ are integrated simultaneously with $\mathbf{z}^i(t) = \mathbf{z}^i(0) + \int_0^t \mathbf{f}(\mathbf{z}^i(\tau), \tau) \mathrm{d}\tau$ with a sample $\mathbf{z}^i(0) \sim \mu$. We summarize pseudo-code for learning ACNF annealed sampler in Algorithm 2. Once ACNF sampler is learned, it can generate unbiased samples: generate one-shot samples from ACNF with flow length $t$ according to computation budget; correct samples by resampling according to importance weights like (Müller et al., 2019) or by Markov Chain Monte Carlo methods with Metropolis-Hastings correction.

# 6 EXPERIMENTS

## 6.1 DENSITY ESTIMATION ON TOY 2D DISTRIBUTIONS

Before we deploy ACNF for modeling complex distributions, we first examine it on a 2-modal Gaussian mixture in 2D and use a standard Gaussian as the base distribution. Figure 4 shows that the potential field of the learned ACNF is very similar to the numerical PDE solutions of eq.(7) while the potential of CNF converges much slower than that of ACNF and then diverges after $T$. See Appendix A.7 for experiment details and comparison on the choices of $\lambda, T$ and other regularization methods.

We then train vanilla CNF, RNODE (Finlay et al., 2020) and ACNFs to model for various 2D toy distributions and visualize the density estimation along flows. Figure 5 shows the densities at $t \in [0, 2T], T = 10$ by learned CNF and ACNFs with various regularization coefficients for 2-moon distribution. The densities that are close to the target distribution are highlighted inside the red border. We show that even slight regularization makes the learned flows to 1) converge much faster towards the target; 2) maintain the best estimations for long time after $T$. Seen from the left of Figure 6, the quantitative evaluation on the log-likelihood estimates implies the same conclusion. More analysis on different $T$ and experiment setups are given in Appendix A.8.

One may suspect that more complex dynamics explain the faster ascent of likelihood estimates. To validate the actual improvements by ACNF, we report the number of function evaluations (NFEs) like (Finlay et al., 2020) by counting the times when a numerical solver calls to evaluate dynamics

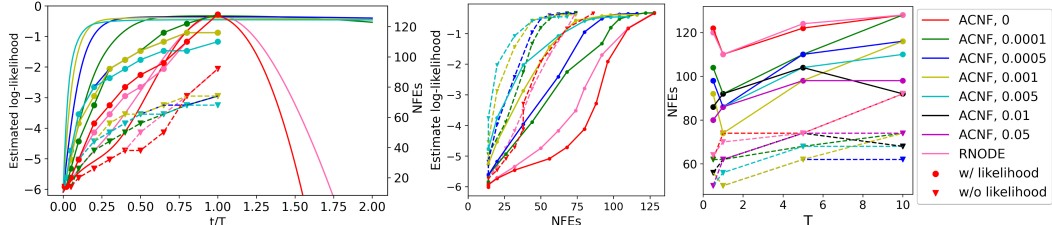

Figure 6: Left: comparison on estimated log-likelihoods of models trained under different regularization $\lambda$ as Figure 5. Middle: log-likelihood vs NFE as the left figure. Right: comparison on NFEs evaluated at $t/T = 1$ of vanilla CNF, RNODE and ACNF trained with various flow length $T$ and $\lambda$.

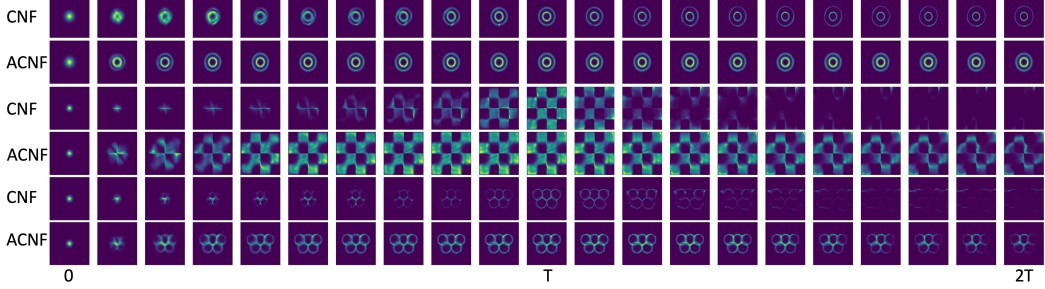

Figure 7: Comparison on density estimations of trained ACNF and vanilla CNF models on various two-dimensional toy distributions along flows with increasing $t \in [0, 2T]$

function in integral, with and without log-likelihood estimate for all models. The marks in the left of Figure 6 show NFEs along flows while the middle one plots log-likelihood estimates versus NFEs. ACNFs clearly demonstrate that they learn even less complex dynamics than CNF and RNODE, and log-likelihood gain per NFE of ACNFs are much higher than the two baselines especially at early stage. Regarding ascent regularization, a larger coefficient leads to more rapid gain on the log-likelihoods initially, however, too large regularization over-constrains models to reach a good maximum. A moderate regularization benefits both maximum likelihood and faster convergence. Furthermore, we report NFEs at $t/T = 1$ for CNF, RNODE and ACNFs trained with various $\lambda = 0.0001, 0.005, 0.001, 0.005, 0.01, 0.05$ for flow length $T = 0.5, 1, 5, 10$ on the right of Figure 6. ACNFs have generally lower NFEs than CNFs and RNODEs, and most models report the lowest NFEs at $T = 1$. It indicates that optimizing $T$ like TO-FLOW (Du et al., 2022) and STEER (Ghosh et al., 2020) may decrease computational cost at $T$, however, neither strategy can accelerate the convergence of flow and prevent deterioration like ACNFs as shown by the density estimate of CNF in Figure 14 in Appendix A.8. Figure 7 shows density evaluations on other multi-modal distributions. Learned ACNFs show faster convergence than CNFs for all distributions and even give a better maximum density estimation on the challenging task, e.g. Olympics distribution.

## 6.2 DENSITY ESTIMATION ON REAL DATASETS

We demonstrate density estimations on real-world benchmarks datasets including POWER, GAS, HEPMASS, MINIBOONE from the UCI machine learning data repository and BSDS300 natural image patches. Like FFJORD, all tabular datasets and BSDS300 are pre-processed as in (Papamakarios et al., 2017). Table 1 reports the averaged NLLs on test data for FFJORD, RNODE and ACNFs trained with different $\lambda$. The detailed description of experiments and models refers to Appendix A.9. Although FFJORD with multi-step flows increases the flexibility of flows, it tends to have a worse performance than the base distribution initially and then improves NLL mainly at the late stage of flows. A larger ascent regularization of ACNFs contributes to more rapid initial increases on NLL that these flows transform the base distribution faster towards the data distribution. When training on HEPMASS and BSDS300, a too large regularization coefficient impedes model to converge.

## 6.3 ACNF AS A FASTER ANNEALING SAMPLING PROPOSAL FOR UNBIASED SAMPLING

Following Algorithm 2, we train CNF and ACNFs with regularization coefficients $\lambda = 0.0001, 0.001, 0.01$ to learn the flow of annealed targets. We evaluate the estimates of $\log Z$ on

Table 1: Averaged negative log-likelihoods (NLLs) on test data for density estimation.

| Datasets | Model | $0.1T$ | $0.25T$ | $0.5T$ | $0.75T$ | $T$ | $t > T$‡ |
|---|---|---|---|---|---|---|---|
| POWER | FFJORD†§ | 7.47 | 5.97 | 4.63 | 2.55 | $-0.42$ | 5.02 |
| | RNODE | 7.80 | 6.11 | 4.80 | 2.64 | $-0.46$ | 4.59 |
| | ACNF, $1e^{-4}$ | 7.03 | 5.02 | 2.63 | 0.81 | $-0.40$ | 0.12 |
| | ACNF, $1e^{-3}$ | 6.26 | 4.15 | 1.99 | **0.48** | $-0.44$ | 0.02 |
| | ACNF, $1e^{-2}$ | **5.43** | **3.47** | **1.87** | **0.48** | **$-0.37$** | **$-0.26$** |
| GAS | FFJORD†§ | 11.76 | 241.13 | 145.75 | 132.12 | $-8.60$ | 279.97 |
| | RNODE | 12.03 | 102.77 | 78.20 | 45.38 | **$-8.79$** | 90.23 |
| | ACNF, $1e^{-4}$ | 7.25 | 3.31 | $-1.28$ | $-5.5$ | $-8.56$ | $-6.88$ |
| | ACNF, $1e^{-3}$ | 5.58 | 1.65 | $-1.73$ | $-5.61$ | $-8.45$ | $-7.63$ |
| | ACNF, $1e^{-2}$ | **3.57** | **$-0.14$** | **$-3.73$** | **$-6.42$** | $-8.57$ | **$-8.33$** |
| HEPMASS | FFJORD†§ | 35.51 | 40.19 | 82.08 | 106.81 | 14.90 | 42.09 |
| | RNODE | 35.02 | 40.59 | 68.91 | 96.47 | 15.34 | 39.09 |
| | ACNF, $1e^{-4}$ | 27.75 | 24.88 | 20.63 | 17.17 | 14.95 | **15.36** |
| | ACNF, $1e^{-3}$ | **25.24** | **22.78** | **18.81** | **15.94** | **14.88** | 15.52 |
| MINIBOONE | FFJORD§ | 58.54 | 53.98 | 42.12 | 24.55 | 10.50 | 16.25 |
| | RNODE | 57.98 | 53.65 | 41.79 | 24.12 | 10.77 | 15.86 |
| | ACNF, $1e^{-4}$ | 54.15 | 43.76 | 29.29 | 19.00 | 10.63 | 13.95 |
| | ACNF, $1e^{-3}$ | 52.64 | 42.87 | 28.92 | 18.72 | **10.40** | 12.57 |
| | ACNF, $1e^{-2}$ | **51.52** | **40.52** | **26.53** | **17.16** | 10.95 | **12.14** |
| BSDS300 | FFJORD†§ | 41.72 | 30.11 | 0.91 | $-85.92$ | $-156.60$ | $-77.20$ |
| | RNODE | 42.38 | 33.29 | 1.74 | $-84.03$ | **$-156.71$** | $-89.35$ |
| | ACNF, $1e^{-4}$ | **35.24** | 24.10 | $-14.75$ | $-100.35$ | $-156.0$ | **$-125.67$** |
| | ACNF, $1e^{-3}$ | 37.45 | **21.53** | **$-17.84$** | **$-105.11$** | $-156.5$ | $-120.89$ |

† FFJORD uses multi-step flow models for some datasets, so the total length of flow is no longer training configuration of $T$ but $T$ times the number of flow steps. $T$ listed here refers the total length of flow. ‡ The flow length $t$ after $T$ is set slightly different among datasets due to the multi-step FFJORD: $1.2T$ for POWER and GAS, $1.1T$ for HEPMASS, and $1.25T$ for MINIBOONE and BSDS300, but it is always the *same* cross different models. §FFJORDs are trained to match the performance as originally reported.

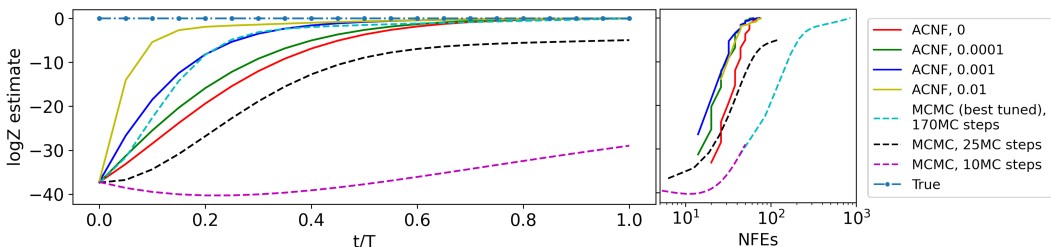

Figure 8: Left: comparison on estimated $\log Z$ by different methods along the flow over 5 different runs. Right: estimated $\log Z$ vs NFEs. See Figure 18 for generated sample comparisons.

a Gaussian mixture target with 8 components whose means are fixed evenly in space and standard deviations as 0.3 and the base distribution is a Gaussian, $\mathcal{N}(\mathbf{0}, 3^2\mathbf{I})$, to give adequate support. Figure 8 compares the estimates evaluated along flows and reports estimates versus NFEs. We benchmark CNF and ACNFs with the linear annealed target $\log \gamma_k(\cdot) = \beta_k \log \gamma(\cdot) + (1 - \beta_k) \log \pi_0$, where scheduling is $\beta_k = k/K = t_k/T$ and $K = 20$, using $\{170, 25, 10\}$-step Metropolis sampler between each intermediate target like (Arbel et al., 2021; Wu et al., 2020). As ACNFs converge faster towards the target than CNF, the estimates via one-shot samples from ACNFs are less biased than that from CNF especially at the initial of flows. Besides, ACNFs are more computationally efficient in terms of accuracy gain per NFEs. ACNFs with $\lambda = 0.01, 0.001$ show less biased estimates earlier than even the best tuned linear annealed target. Besides, the linear annealed target requires at least 1 order more computations than ACNFs for comparable accuracy due to slow mixing of Metropolis sampler, and its performance is very sensitive to the number of MC steps. Figure 18 in Appendix A.10 shows the generated samples by all methods in Figure 8. Adding MC steps with learned ACNFs can further accelerate sample convergence and increase the expressiveness of flows.

Table 2: Averaged negative ELBO on MNIST datasets under different length of flows $t$.

| Model | $0.1T$ | $0.25T$ | $0.5T$ | $0.75T$ | $T$ | $1.2T$ |
|---|---|---|---|---|---|---|
| VAE-FFJORD | 85.90 | 85.07 | 83.96 | 83.26 | 82.88(82.82†) | 89.74 |
| VAE-ACNF, $1e^{-4}$ | 85.62 | 84.60 | 83.45 | 83.06 | **82.74** | 85.67 |
| VAE-ACNF, $1e^{-3}$ | **84.70** | **83.95** | **83.22** | **82.53** | 82.80 | **84.37** |

† originally reported in FFJORD

Figure 9: Reconstructions from VAE-ACNF and VAE and original data for some challenging samples.

## 6.4 VARIATIONAL INFERENCE WITH ACNFS

In addition to density estimation and unbiased sampling, CNF provides more flexible variational approximation to improve variational inference (Rezende and Mohamed, 2015). We follow the experiment setup as (Grathwohl et al., 2018), that uses an encoder/decoder with a CNF architecture, of which the encoder gives the base latent posterior approximation for CNF and the decoder decodes latent inference at the end of the flow back to observation dimension. To train a VAE-ACNF model, the log weight in eq.(10) is replaced by an ELBO estimate like (Kingma and Welling, 2014).

We evaluate VAE-ACNF to VAE-FFJORD and vanilla VAE without flow on MNIST data. To make a fair comparison, we fix the learned encoder-decoder when training all flows. A detailed description of model architecture and experimental setup can be found in Appendix A.11. The averaged negative ELBO on test data of VAE is 86.50 and Table 2 reports that of VAE-FFJORD and VAE-ACNFs with $\lambda = 1e^{-4}, 1e^{-3}$ along the flows. Compared to VAE-FFJORD, VAE-ACNFs show faster descent on negative ELBO at the initial of the flows, and a larger coefficient shows faster convergence of the variational approximation. VAE-ACNFs also circumvent the flow deterioration by VAE-FFJORD, thanks to the self-stabilization behavior of ACNF. Figure 9 and Figure 19 in Appendix A.11 show some reconstruction examples from VAE-ACNF. These reconstructions tend to correct some defects in original images, add details to strengthen identities while remaining sharp.

## 7 SCOPES AND LIMITATIONS

While we have demonstrated that ascent regularization is effective to learn flows that converge faster to target distributions, there are still a number of limitations, which we would like to address in the future. First, more efficient implementations on score function evaluation e.g. by estimators or model design or learning via score matching (Song et al., 2021) can accelerate training for high-dimensional problems. Second, Hypernet (Ha et al., 2016) used in experiments is found suitable to illustrate faster convergence behavior of ACNF as time exerts a large impact on the dynamics, however, it is slower to train than other simpler network architectures. A better architecture may improve the training speed while maintaining the desired characteristics of flows. Third, although the proposed ACNF and ascent regularization have been discussed under the framework of CNF, the concept can be easily extended and explored for score-based models and other stochastic flows. Finally, ACNF and ascent regularization can be applied for a sequence of distributions, e.g. inference of sequential data.

## 8 CONCLUSION

We introduce ACNFs, a new class of CNFs, that define flows with monotonic convergence toward a target distribution. We derive the dynamics for the steepest ACNF and propose a practical implementation to learn parametric ACNFs via ascent regularization. We demonstrate ACNF in three use cases: modeling data and performing density estimation, learning an annealed sampler for unbiased sampling, and learning variational approximation for variational inference. The learned ACNFs illustrate three beneficial behaviors: 1) faster convergence to the target distribution with less computation; 2) self-stabilization to mitigate performance deterioration; 3) insensitivity to flow training length $T$. Experiments on both toy distributions and real-world datasets demonstrate the effectiveness of ascent regularization on learning ACNFs for various purposes.

ACKNOWLEDGMENTS

This work is supported by the Wallenberg AI, Autonomous Systems and Software Program (WASP). Y. K. is a member of the ELLIIT Strategic Research Area at Lund University.

ETHICS STATEMENT

As this work mainly concerns to propose a flow-based model and practical implementation for learning, it does not involve human subjects, practices to data set releases, or security and privacy issue. At this stage of study, we do not foresee the effects of potential system failures due to weaknesses in the proposed methods.

REPRODUCIBILITY STATEMENT

All proposition (proposition 1) and theorems (theorem 1 and 2) proposed in this paper are proved with details in Appendix A.1, A.2 and A.5 as well as other minor derivations mentioned in main body of the paper. The pseudo-code for both learning cases are provided in Algorithm 1 and Algorithm 2. The datasets, models, experiment setups for each demonstration are described in details in Appendix A.8 $\sim$ A.11. Furthermore, we attach some source codes in supplementary material for further checkup.

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

# A APPENDIX

## A.1 PROOF OF THE INSTANTANEOUS CHANGE OF LOG-LIKELIHOOD ESTIMATE

**Proposition** (Instantaneous Change of Log-likelihood Estimate). *Let $\mathbf{z}(t)$ be a finite continuous random variable at time $t$ as the solution of a differential equation $\frac{d\mathbf{z}(t)}{dt} = \mathbf{f}(\mathbf{z}(t), t)$ with initial value $\mathbf{z}(0) = \mathbf{x}$. Assuming that $\tilde{p}_0 = \mu$ at $t = 0$ and $\mathbf{f}$ is uniformly Lipschitz continuous in $\mathbf{z}$ and $t$, then the change in estimated log-likelihood $\log \tilde{p}_t(\mathbf{x})$ at $t$ follows a differential equation:*

$$\frac{d \log \tilde{p}_t(\mathbf{x})}{dt} = \boldsymbol{\nabla} \cdot \mathbf{f}(\mathbf{z}(t), t) + \nabla \log \mu(\mathbf{z}(t)) \cdot \mathbf{f}(\mathbf{z}(t), t).$$

*Proof.* To prove this theorem, we take the infinitesimal limit of finite changes of $\log \tilde{p}_t(\mathbf{x})$ through time. As $\mathbf{f}$ is assumed to be Lipschitz continuous in $\mathbf{z}(t)$ and $t$, $\Phi_t(\mathbf{x})$ represents the unique solution of the ODE (eq.(2)) at time $t$ on the initial value $\mathbf{x}$:

$$\Phi_t(\mathbf{x}) = \mathbf{x} + \int_0^t \mathbf{f}(\Phi_\tau(\mathbf{x}), \tau) d\tau,$$

We also denote the transformation on $\mathbf{z}(t)$ over an $\epsilon$ change in time as:

$$\mathbf{z}(t + \epsilon) = \Phi_\epsilon(\mathbf{z}(t)) = \Phi_{t+\epsilon}(\mathbf{x}).$$

Using the definition of estimated log density $\tilde{p}_t(\mathbf{x})$ in eq.(4), the infinitesimal limit is:

$$\frac{d \log \tilde{p}_t(\mathbf{x})}{dt} := \lim_{\epsilon \to 0^+} \frac{1}{\epsilon} \left( \log\left|\det\left(\mathcal{J}_{\boldsymbol{\Phi}_{t+\epsilon}}(\mathbf{x})\right)\right| - \log\left|\det\left(\mathcal{J}_{\boldsymbol{\Phi}_t}(\mathbf{x})\right)\right| + \log \mu(\boldsymbol{\Phi}_{t+\epsilon}(\mathbf{x})) - \log \mu(\boldsymbol{\Phi}_t(\mathbf{x})) \right)$$

$$= \lim_{\epsilon \to 0^+} \frac{1}{\epsilon} \left( \log\left|\det\left(\mathcal{J}_{\boldsymbol{\Phi}_{t+\epsilon}}(\mathbf{x})\right)\right| - \log\left|\det\left(\mathcal{J}_{\boldsymbol{\Phi}_t}(\mathbf{x})\right)\right| \right) + \lim_{\epsilon \to 0^+} \frac{1}{\epsilon} \left( \log \mu(\boldsymbol{\Phi}_{t+\epsilon}(\mathbf{x})) - \log \mu(\boldsymbol{\Phi}_t(\mathbf{x})) \right).$$

The derivation of first term is very similar to (Chen et al., 2018, theorem 1) except the sign of the function:

$$\lim_{\epsilon \to 0^+} \frac{1}{\epsilon} \left( \log\left|\det\left(\mathcal{J}_{\boldsymbol{\Phi}_{t+\epsilon}}(\mathbf{x})\right)\right| - \log\left|\det\left(\mathcal{J}_{\boldsymbol{\Phi}_t}(\mathbf{x})\right)\right| \right)$$

$$= \lim_{\epsilon \to 0^+} \frac{1}{\epsilon} \left( \log\left|\det\left(\mathcal{J}_{\boldsymbol{\Phi}_\epsilon}(\mathbf{z}(t))\right)\right| + \log\left|\det\left(\mathcal{J}_{\boldsymbol{\Phi}_t}(\mathbf{x})\right)\right| - \log\left|\det\left(\mathcal{J}_{\boldsymbol{\Phi}_t}(\mathbf{x})\right)\right| \right)$$

$$= \lim_{\epsilon \to 0^+} \frac{1}{\epsilon} \log\left|\det\left(\mathcal{J}_{\boldsymbol{\Phi}_\epsilon}(\mathbf{z}(t))\right)\right|,$$

where we summarize the main steps:

$$\lim_{\epsilon \to 0^+} \frac{\log |\det \mathcal{J}_{\boldsymbol{\Phi}_\epsilon}(\mathbf{z}(t))|}{\epsilon} = \lim_{\epsilon \to 0^+} \frac{\frac{\partial}{\partial \epsilon} \log |\det \mathcal{J}_{\boldsymbol{\Phi}_\epsilon}(\mathbf{z}(t))|}{\frac{\partial \epsilon}{\partial \epsilon} \to 1} \qquad \text{(L'Hopital's rule)}$$

$$= \lim_{\epsilon \to 0^+} \frac{\frac{\partial}{\partial \epsilon} |\det \mathcal{J}_{\boldsymbol{\Phi}_\epsilon}(\mathbf{z}(t))|}{|\det \mathcal{J}_{\boldsymbol{\Phi}_\epsilon}(\mathbf{z}(t))| \to 1}$$

$$= \lim_{\epsilon \to 0^+} \frac{\partial}{\partial \epsilon} |\det \mathcal{J}_{\boldsymbol{\Phi}_\epsilon}(\mathbf{z}(t))|$$

$$= \lim_{\epsilon \to 0^+} \mathrm{Tr}\left( \mathrm{adj}\left( \frac{\partial}{\partial \mathbf{z}(t)} \boldsymbol{\Phi}_\epsilon(\mathbf{z}(t)) \right) \frac{\partial}{\partial \epsilon} \frac{\partial}{\partial \mathbf{z}(t)} \boldsymbol{\Phi}_\epsilon(\mathbf{z}(t)) \right) \qquad \text{(Jacobi's formula)}$$

$$= \mathrm{Tr}\left( \lim_{\epsilon \to 0^+} \frac{\partial}{\partial \epsilon} \frac{\partial}{\partial \mathbf{z}(t)} \boldsymbol{\Phi}_\epsilon(\mathbf{z}(t)) \right) \qquad \text{(adjacent matrix} \to \mathbf{I} \text{ as } \epsilon \to 0^+)$$

$$= \mathrm{Tr}\left( \lim_{\epsilon \to 0^+} \frac{\partial}{\partial \epsilon} \frac{\partial}{\partial \mathbf{z}(t)} \left( \mathbf{z}(t) + \epsilon \mathbf{f}(\mathbf{z}(t), t) + o(\epsilon^2) + \dots \right) \right)$$

$$= \mathrm{Tr}\left( \lim_{\epsilon \to 0^+} \frac{\partial}{\partial \epsilon} \left( \mathbf{I} + \epsilon \frac{\partial \mathbf{f}(\mathbf{z}(t), t)}{\partial \mathbf{z}(t)} + o(\epsilon^2) + \dots \right) \right)$$

$$= \mathrm{Tr}\left( \lim_{\epsilon \to 0^+} \left( \frac{\partial \mathbf{f}(\mathbf{z}(t), t)}{\partial \mathbf{z}(t)} + o(\epsilon) + \dots \right) \right)$$

$$= \boldsymbol{\nabla} \cdot \mathbf{f}(\mathbf{z}(t), t).$$

Before deriving the second term, we take the first-order Taylor expansion of $\log \mu(\boldsymbol{\Phi}_\epsilon(\mathbf{z}(t)))$ at $\mathbf{z}(t) = \boldsymbol{\Phi}_t(\mathbf{x})$:

$$\log \mu(\boldsymbol{\Phi}_{t+\epsilon}(\mathbf{x})) = \log \mu(\boldsymbol{\Phi}_\epsilon(\mathbf{z}(t))) = \log \mu(\mathbf{z}(t)) + \nabla \log \mu(\mathbf{z}(t)) \cdot (\boldsymbol{\Phi}_\epsilon(\mathbf{z}(t)) - \mathbf{z}(t)) + o(\epsilon^2) + \dots,$$

hence,

$$
\begin{aligned}
& \lim_{\epsilon \to 0^+} \frac{\log \mu(\boldsymbol{\Phi}_{t+\epsilon}(\mathbf{x})) - \log \mu(\boldsymbol{\Phi}_t(\mathbf{x}))}{\epsilon} \\
=& \lim_{\epsilon \to 0^+} \frac{\log \mu(\mathbf{z}(t)) + \nabla \log \mu(\mathbf{z}(t)) \cdot (\boldsymbol{\Phi}_\epsilon(\mathbf{z}(t)) - \mathbf{z}(t)) + o(\epsilon^2) + \dots - \log \mu(\mathbf{z}(t))}{\epsilon} \\
=& \lim_{\epsilon \to 0^+} \nabla \log \mu(\mathbf{z}(t)) \cdot \frac{\boldsymbol{\Phi}_\epsilon(\mathbf{z}(t)) - \mathbf{z}(t)}{\epsilon} + o(\epsilon) + \dots \\
=& \nabla \log \mu(\mathbf{z}(t)) \cdot \mathbf{f}(\mathbf{z}(t), t).
\end{aligned}
$$

Therefore, the differential of $\log \tilde{p}_t(\mathbf{x})$ is:

$$\frac{\mathrm{d} \log \tilde{p}_t(\mathbf{x})}{\mathrm{d}t} = \boldsymbol{\nabla} \cdot \mathbf{f}(\mathbf{z}(t), t) + \nabla \log \mu(\mathbf{z}(t)) \cdot \mathbf{f}(\mathbf{z}(t), t).$$

$\square$

To show the relation between two differentials $\frac{\mathrm{d} \log \tilde{p}_t(\mathbf{x})}{\mathrm{d}t}$ and $\frac{\mathrm{d} \log p_t(\mathbf{z}(t))}{\mathrm{d}t}$, we first need the relation between $\log \tilde{p}_t(\mathbf{x})$ and $\log p_t(\mathbf{z}(t))$:

$$\log \tilde{p}_t(\mathbf{x}) = \log p(\mathbf{x}) + \log \mu(\mathbf{z}(t)) - \log p_t(\mathbf{z}(t)).$$

Taking the total derivative on both l.h.s. and r.h.s. of last equation:

$$
\begin{aligned}
\frac{\mathrm{d} \log \tilde{p}_t(\mathbf{x})}{\mathrm{d}t} =& \frac{\mathrm{d} \log \mu(\mathbf{z}(t))}{\mathrm{d}t} - \frac{\mathrm{d} \log p_t(\mathbf{z}(t))}{\mathrm{d}t} \\
=& \nabla \log \mu(\mathbf{z}(t)) \cdot \mathbf{f}(\mathbf{z}(t), t) - \frac{\mathrm{d} \log p_t(\mathbf{z}(t))}{\mathrm{d}t} \\
=& \nabla \log \mu(\mathbf{z}(t)) \cdot \mathbf{f}(\mathbf{z}(t), t) + \boldsymbol{\nabla} \cdot \mathbf{f}(\mathbf{z}(t), t).
\end{aligned}
$$

The total derivative $\frac{\mathrm{d} \log \tilde{p}_t(\mathbf{x})}{\mathrm{d}t}$ is defined on the fixed variable $\mathbf{x}$, while the infinitesimal change on r.h.s. is evaluated on the variable $\mathbf{z}(t)$. So solving $\log \tilde{p}_t(\mathbf{x})$ requires to simulate $\mathbf{z}(t)$ simultaneously. Different to solving $\log p_t(\mathbf{z}(t))$ on the reversed direction of solving $\mathbf{z}(t)$, $\log \tilde{p}_t(\mathbf{x})$ only needs the trajectory of $\mathbf{z}(\tau), \tau \in [0, t]$, while $\log p_t(\mathbf{z}(t))$ requires to know the whole trajectory of $\mathbf{z}(\tau), \tau \in [0, T]$. Therefore, using $\log \tilde{p}_t(\mathbf{x})$ is more advantageous when evaluating models at any $t$ other than $T$ or at multiple $t$.

As for training, since $p_T$ is specified as $\mu$ at $T$, maximizing $\log p(\mathbf{x})$ in vanilla CNF is essentially equivalent to maximizing $\log \tilde{p}_t(\mathbf{x})$ in ACNF.

If we take the time partial derivative on the log-likelihood equation, then

$$\frac{\partial \log \tilde{p}_t(\mathbf{x})}{\partial t} = \frac{\partial \log \mu(\mathbf{z}(t))}{\partial t} - \frac{\partial \log p_t(\mathbf{z}(t))}{\partial t} = -\frac{\partial \log p_t(\mathbf{z}(t))}{\partial t},$$

so that the convergence rate of distribution estimate $\tilde{p}_t(\mathbf{x})$ towards $p(\mathbf{x})$ is equivalent to the normalized distribution $p_t(\mathbf{z})$ towards $\mu(\mathbf{z})$.

## A.2 PROOF OF THE DYNAMICS FOR THE STEEPEST ASCENT CONTINUOUS NORMALIZING FLOWS

**Theorem** (Dynamics for Steepest Ascent Continuous Normalizing Flows). *Let $\mathbf{z}(t)$ be a finite continuous random variable and the solution of a differential equation $\frac{\mathrm{d}\mathbf{z}(t)}{\mathrm{d}t} = \mathbf{f}(\mathbf{z}(t), t)$ with initial*

value $\mathbf{z}(0) = \mathbf{x}$. *Its probability $p_t(\mathbf{z}(t))$ subjects to the continuity equation $\partial_t p_t + \boldsymbol{\nabla} \cdot (p_t \mathbf{f}) = 0$. The dynamics of the steepest flow for decreasing $KL(p_t(\mathbf{z}(t))||\mu(\mathbf{z}(t)))$ is*

$$\mathbf{f}^*(\mathbf{z}(t), t) = \nabla \log \mu(\mathbf{z}(t)) - \frac{\nabla p_t(\mathbf{z}(t))}{p_t(\mathbf{z}(t))} = \nabla \log \mu(\mathbf{z}(t)) - \nabla \log p_t(\mathbf{z}(t)).$$

To keep this proof simple, we derive this theorem in Euclidean space. If readers are familiar with non-Euclidean metric spaces, we refer more rigid of Wasserstein gradient flow proof in (Ambrosio et al., 2005).

*Proof.* Assuming that $N$ samples $X = \{\mathbf{x}_i\}_{i=1:N} \in \mathbb{R}^{Nd}$ are drawn from $p(\mathbf{x})$, the averaged negative estimated log-likelihood at time $t$ is:

$$J(\boldsymbol{\Phi}_t) = -\frac{1}{N} \sum_{i=1}^N \log \tilde{p}_t(\mathbf{x}_i) = \frac{1}{N} \sum_{i=1}^N \left( \log p_t(\boldsymbol{\Phi}_t(\mathbf{x}_i)) - \log \mu(\boldsymbol{\Phi}_t(\mathbf{x}_i)) - \log p(\mathbf{x}_i) \right).$$

Using the chain rule, the derivative of $J(\boldsymbol{\Phi}_t)$ w.r.t. $\boldsymbol{\Phi}_t(\mathbf{x}_i)$ is:

$$[\nabla J(\boldsymbol{\Phi}_t)]_i = \nabla \log p_t(\boldsymbol{\Phi}_t(\mathbf{x}_i)) - \nabla \log \mu(\boldsymbol{\Phi}_t(\mathbf{x}_i)),$$

where $\nabla J(\boldsymbol{\Phi}_t)$ is a matrix that each row is for each sample $i = 1, 2, \ldots, N$ and each column is for each dimension $j = 1, 2, \ldots, d$.

To numerically compute the solutions of Euler-Lagrange equation, i.e. $\nabla J(\boldsymbol{\Phi}_t) = \mathbf{0}$, we use *gradient descent* to define the evolution of transformation $\boldsymbol{\Phi}_t$ for each $\mathbf{x}_i$:

$$\frac{\mathrm{d}\boldsymbol{\Phi}_t(\mathbf{x}_i)}{\mathrm{d}t} = -[\nabla J(\boldsymbol{\Phi}_t)]_i = \nabla \log \mu(\boldsymbol{\Phi}_t(\mathbf{x}_i)) - \nabla \log p_t(\boldsymbol{\Phi}_t(\mathbf{x}_i)),$$

which evolves $\boldsymbol{\Phi}$ in the direction that decreases $J(\boldsymbol{\Phi})$ most rapidly, starting at initial $\boldsymbol{\Phi}_0(\mathbf{x}_i) = \mathbf{x}_i$.

The next step is to extend the assumption of the finite number of data samples $N$ to infinity, i.e. $N \to \infty$, therefore, the objective $J(\boldsymbol{\Phi}_t)$ at time $t$ is updated as:

$$\begin{aligned} J(\boldsymbol{\Phi}_t) &= -\int_U \log \tilde{p}_t(\mathbf{x}) \mathrm{d}\mathbf{x} \\ &= \int_U \left( \log p_t(\boldsymbol{\Phi}_t(\mathbf{x})) - \log \mu(\boldsymbol{\Phi}_t(\mathbf{x})) - \log p(\mathbf{x}) \right) \mathrm{d}\mathbf{x} \\ &= \int_U L(\mathbf{x}, \boldsymbol{\Phi}_t(\mathbf{x}), \nabla \boldsymbol{\Phi}_t(\mathbf{x})) \mathrm{d}\mathbf{x}, \end{aligned}$$

where $\mathbf{x} \in U \subseteq \mathbb{R}^d$ and $L(\mathbf{x}, \boldsymbol{\Phi}_t(\mathbf{x}), \nabla \boldsymbol{\Phi}_t(\mathbf{x})) = \log p_t(\boldsymbol{\Phi}_t(\mathbf{x})) - \log \mu(\boldsymbol{\Phi}_t(\mathbf{x})) - \log p(\mathbf{x})$. For each $j$ dimension of $\boldsymbol{\Phi}_t$, the functional derivative of $J(\boldsymbol{\Phi}_t)$ w.r.t. $[\boldsymbol{\Phi}_t]_j$ is:

$$\begin{aligned} \frac{\delta J(\boldsymbol{\Phi}_t)}{\delta[\boldsymbol{\Phi}_t]_j} &= \frac{\partial L}{\partial[\boldsymbol{\Phi}_t]_j}(\mathbf{x}, \boldsymbol{\Phi}_t(\mathbf{x}), \nabla \boldsymbol{\Phi}_t(\mathbf{x})) - \boldsymbol{\nabla} \cdot \left( \frac{\partial L}{\partial \nabla[\boldsymbol{\Phi}_t]_j}(\mathbf{x}, \boldsymbol{\Phi}_t(\mathbf{x}), \nabla \boldsymbol{\Phi}_t(\mathbf{x})) \right) \\ &= [\nabla \log p_t(\boldsymbol{\Phi}_t(\mathbf{x}))]_j - [\nabla \log \mu(\boldsymbol{\Phi}_t(\mathbf{x}))]_j, \end{aligned}$$

as $\frac{\partial L}{\partial \nabla[\boldsymbol{\Phi}_t]_j} = \mathbf{0}$. Therefore, the gradient descent that defines the evolution of transformation $\boldsymbol{\Phi}_t$ is:

$$\frac{\mathrm{d}\boldsymbol{\Phi}_t(\mathbf{x})}{\mathrm{d}t} = -\frac{\delta J(\boldsymbol{\Phi}_t)}{\delta \boldsymbol{\Phi}_t} = \nabla \log \mu(\boldsymbol{\Phi}_t(\mathbf{x})) - \nabla \log p_t(\boldsymbol{\Phi}_t(\mathbf{x})), \tag{11}$$

therefore, the dynamics for the steepest ascent continuous normalizing flow is:

$$\mathbf{f}^*(\mathbf{z}(t), t) = \frac{\mathrm{d}\boldsymbol{\Phi}_t(\mathbf{x})}{\mathrm{d}t} = \nabla \log \mu(\mathbf{z}(t)) - \nabla \log p_t(\mathbf{z}(t)).$$

$\square$

### A.3 CONVERGENCE RATE OF OPTIMAL ASCENT CONTINUOUS NORMALIZING FLOWS AND ITS RELATION TO LANGEVIN DYNAMICS

The convergence rate of KL divergence w.r.t. $t$ can be derived as (we start from a general flow dynamics $\mathbf{f}$):

$$
\begin{aligned}
\frac{\partial}{\partial t}\mathrm{KL}(p_t(\mathbf{z})\|\mu(\mathbf{z})) &= \frac{\partial}{\partial t}\mathrm{KL}(p(\mathbf{x})\|\tilde{p}_t(\mathbf{x})) \\
&= -\int p(\mathbf{x})\frac{\partial}{\partial t}\tilde{p}_t(\mathbf{x})\mathrm{d}\mathbf{x} \\
&= -\int p(\mathbf{x})\left(\boldsymbol{\nabla}\cdot\mathbf{f}(\mathbf{z}(t),t)+\nabla\log\mu(\mathbf{z}(t))\cdot\mathbf{f}(\mathbf{z}(t),t)\right)\mathrm{d}\mathbf{x} \\
&= -\int p_t(\mathbf{z}(t))\left(\boldsymbol{\nabla}\cdot\mathbf{f}(\mathbf{z}(t),t)+\nabla\log\mu(\mathbf{z}(t))\cdot\mathbf{f}(\mathbf{z}(t),t)\right)\mathrm{d}\mathbf{z}(t) \\
&= -\int p_t(\mathbf{z})\left(\sum_i\frac{\partial f_i(\mathbf{z},t)}{\partial z_i}+\sum_i\frac{\partial\log\mu(\mathbf{z})}{\partial z_i}f_i(\mathbf{z})\right)\mathrm{d}\mathbf{z} \\
&= -\sum_i\left[-f_i(\mathbf{z},t)\frac{\partial p_t(\mathbf{z})}{\partial z_i}+\frac{\partial\log\mu(\mathbf{z})}{\partial z_i}f_i(\mathbf{z})\right] \\
&= -\sum_i\int p_t(\mathbf{z})(-\frac{\partial\log p_t(\mathbf{z})}{\partial z_i}+\frac{\partial\log\mu(\mathbf{z})}{\partial z_i})f_i(\mathbf{z},t)\mathrm{d}\mathbf{z} \\
&= -\mathbb{E}_{p_t}\left[(\nabla\log\mu(\mathbf{z})-\nabla\log p_t(\mathbf{z}))\cdot\mathbf{f}(\mathbf{z},t)\right].
\end{aligned}
\tag{12}
$$

(Liu, 2017)[theorem 3.1] shows similar derivation from discrete transformation perspective and links to Stein variational gradient flows.

When dynamics $\mathbf{f}$ is equal to the fastest flow dynamics $\mathbf{f}^*$ as eq.(6), then the convergence rate becomes negative Fisher divergence (estimated w.r.t. $p_t$):

$$
\frac{\partial}{\partial t}\mathrm{KL}(p_t(\mathbf{z})\|\mu(\mathbf{z})) = -\mathbb{E}_{p_t}\|\nabla\log p_t(\mathbf{z})-\nabla\log\mu(\mathbf{z})\|_2^2.
$$

This convergence rate can be easily proved the same to overdamped Langevin diffusion dynamics which is defined via a stochastic differential equation at case of $\beta = 1$:

$$
\mathrm{d}\mathbf{z}(t) = \nabla\log\mu(\mathbf{z}(t))\mathrm{d}t + \sqrt{2\beta^{-1}}\mathrm{d}\mathbf{W}_t,
\tag{13}
$$

where $\mathbf{W}_t$ is a Brownian motion. Under the Langevin dynamics, the transformed distribution has a PDE:

$$
\begin{aligned}
\frac{\partial p_t(\mathbf{z})}{\partial t} &= -\boldsymbol{\nabla}\cdot(p_t(\mathbf{z})\nabla\log\mu(\mathbf{z})) + \beta^{-1}\Delta p_t(\mathbf{z}) \\
&= -\boldsymbol{\nabla}\cdot(p_t(\mathbf{z})\nabla\log\mu(\mathbf{z})) + \beta^{-1}\boldsymbol{\nabla}\cdot(\nabla p_t(\mathbf{z})) \\
&= -\boldsymbol{\nabla}\cdot(p_t(\mathbf{z})(\nabla\log\mu(\mathbf{z})-\beta^{-1}\nabla\log p_t(\mathbf{z}))).
\end{aligned}
$$

The last line reveals the steepest gradient flow dynamics as eq.(6) when $\beta = 1$.

Therefore, the optimal ascent continuous normalizing flows and overdamped Langevin dynamics transform a distribution equivalently when $\beta = 1$. And this Fokker Plank equation is a linear (w.r.t. $p_t(\mathbf{z})$) and deterministic although Langevin dynamics is stochastic. The main difference between these two flows is that the dynamics of (optimal) ascent continuous normalizing flow is deterministic, so as any particular sample trajectory; while Langevin dynamics defines a stochastic process and sample trajectories are stochastic.

### A.4 DERIVATION OF POTENTIAL FIELD PDE

The optimal dynamics defined in eq.(6) can be rewritten in terms of the potential function $V(\mathbf{z}(t),t)$, as $V(\mathbf{z},t) := \frac{p_t(\mathbf{z})}{\mu(\mathbf{z})}$:

$$
\mathbf{f}^* = \nabla\log\mu(\mathbf{z}(t)) - \nabla\log p(\mathbf{z}(t),t) = -\nabla\log V(\mathbf{z}(t),t).
\tag{14}
$$

The continuity equation reveals the time derivative of the transformed density $p(\mathbf{z}(t), t)$ at $t$:

$$
\begin{aligned}
\frac{\partial p_t(\mathbf{z}(t))}{\partial t} &= -\boldsymbol{\nabla} \cdot (p_t(\mathbf{z}(t))\mathbf{f}(\mathbf{z}(t), t)) \\
&= -p_t(\mathbf{z}(t))\boldsymbol{\nabla} \cdot \mathbf{f}(\mathbf{z}(t), t) - \nabla p_t(\mathbf{z}(t)) \cdot \mathbf{f}(\mathbf{z}(t), t).
\end{aligned}
$$

Therefore, the time derivative of $\log p_t(\mathbf{z})$ with dynamics defined in eq.(14) is:

$$
\begin{aligned}
\frac{\partial \log p_t(\mathbf{z}(t))}{\partial t} = \frac{1}{p_t(\mathbf{z}(t))}\frac{\partial p_t(\mathbf{z}(t))}{\partial t} &= -\boldsymbol{\nabla} \cdot \mathbf{f}(\mathbf{z}(t), t) - \nabla \log p_t(\mathbf{z}(t)) \cdot \mathbf{f}(\mathbf{z}(t), t) \\
&= \Delta \log V(\mathbf{z}(t), t) + \nabla \log p_t(\mathbf{z}(t)) \cdot \nabla \log V(\mathbf{z}(t), t).
\end{aligned}
$$

Using the last equation, the time derivative of $\log V(\mathbf{z}, t)$ is derived as:

$$
\begin{aligned}
\frac{\partial \log V(\mathbf{z}, t)}{\partial t} :=& \frac{\partial \log p_t(\mathbf{z}(t))}{\partial t} - \frac{\partial \log \mu(\mathbf{z}(t))}{\partial t} \\
=& -\boldsymbol{\nabla} \cdot \mathbf{f}(\mathbf{z}(t), t) - \nabla \log p_t(\mathbf{z}(t)) \cdot \mathbf{f}(\mathbf{z}(t), t) - \nabla \log \mu(\mathbf{z}(t)) \cdot \mathbf{f}(\mathbf{z}(t), t) \\
=& \Delta \log V(\mathbf{z}(t), t) + (\nabla \log p_t(\mathbf{z}(t)) + \nabla \log \mu(\mathbf{z}(t))) \cdot \nabla \log V(\mathbf{z}(t), t) \\
=& \Delta \log V(\mathbf{z}(t), t) + (2\nabla \log \mu(\mathbf{z}(t)) + \nabla \log V(\mathbf{z}(t), t)) \cdot \nabla \log V(\mathbf{z}(t), t),
\end{aligned}
$$

therefore, the time derivative of potential field is:

$$
\frac{\partial V(\mathbf{z}, t)}{\partial t} = \Delta V(\mathbf{z}, t) + 2\nabla \log \mu(\mathbf{z}) \cdot \nabla V(\mathbf{z}, t) + \nabla \log V(\mathbf{z}, t) \cdot \nabla V(\mathbf{z}, t). \tag{15}
$$

When $t = 0$, $V(\mathbf{x}, 0) = \frac{p(\mathbf{x})}{\mu(\mathbf{x})}$; when $t \to \infty$, $V(\mathbf{z}, t) \equiv 1, \forall \mathbf{z}$.

## A.5 INSTANTANEOUS CHANGE OF SCORE FUNCTION

**Theorem** (Instantaneous Change of Score Function). *Let $\mathbf{z}(t)$ be a finite continuous random variable with probability density $p_t(\mathbf{z}(t))$ at time $t$. Let $\frac{d\mathbf{z}(t)}{dt} = \mathbf{f}(\mathbf{z}(t), t)$ be a differential equation describing a continuous-in-time transformation of $\mathbf{z}(t)$. Assuming that $\mathbf{f}$ is uniformly Lipschitz continuous in $\mathbf{z}$ and $t$, the infinitesimal change in the gradient of log-density at $t$ is*

$$
\frac{d\nabla \log p_t(\mathbf{z}(t))}{dt} = -\nabla \log p_t(\mathbf{z}(t))\frac{\partial \mathbf{f}(\mathbf{z}(t), t)}{\partial \mathbf{z}(t)} - \nabla\left(\boldsymbol{\nabla} \cdot \mathbf{f}(\mathbf{z}(t), t)\right).
$$

*Proof.* As $\mathbf{f}$ is assumed to be Lipschitz continuous in $\mathbf{z}(t)$ and $t$, $\Phi_t(\mathbf{x})$ represents the unique solution map. We denote the transformation on $\mathbf{z}(t + \epsilon)$ reversed over an $\epsilon$ change in time as:

$$
\mathbf{z}(t + \epsilon) = \boldsymbol{\Phi}_\epsilon(\mathbf{z}(t)), \ \mathbf{z}(t) = \boldsymbol{\Phi}_{-\epsilon}\left(\mathbf{z}(t + \epsilon)\right),
$$

and applying the change of variable theorem on $\log p_{t+\epsilon}(\mathbf{z}(t + \epsilon))$, defined on the variable $\mathbf{z}(t + \epsilon)$:

$$
\begin{aligned}
\log p_{t+\epsilon}(\mathbf{z}(t + \epsilon)) &= \log p_t(\mathbf{z}(t)) - \log|\det \mathcal{J}_{\boldsymbol{\Phi}_\epsilon}(\mathbf{z}(t))| \\
&= \log p_t(\Phi_{-\epsilon}(\mathbf{z}(t + \epsilon))) - \log|\det \mathcal{J}_{\boldsymbol{\Phi}_\epsilon}(\boldsymbol{\Phi}_{-\epsilon}(\mathbf{z}(t + \epsilon)))|.
\end{aligned}
$$

Taking the derivative of $\log p_{t+\epsilon}(\mathbf{z}(t + \epsilon))$ w.r.t. $\mathbf{z}(t + \epsilon)$ on both l.h.s. and r.h.s. of the last equation:

$$
\nabla \log p_{t+\epsilon}(\mathbf{z}(t + \epsilon)) = (\nabla \log p_t(\mathbf{z}(t)) - \nabla \log|\det \mathcal{J}_{\boldsymbol{\Phi}_\epsilon}(\mathbf{z}(t))|)\frac{\partial \boldsymbol{\Phi}_{-\epsilon}(\mathbf{z}(t + \epsilon))}{\partial \mathbf{z}(t + \epsilon)},
$$

and the infinitesimal limit of finite changes of gradient of log density can be defined:

$$
\frac{\mathrm{d}\nabla \log p_t(\mathbf{z}(t))}{\mathrm{d}t} := \lim_{\epsilon \to 0^+} \frac{1}{\epsilon} \left( \nabla \log p_{t+\epsilon}(\mathbf{z}(t+\epsilon)) - \nabla \log p_t(\mathbf{z}(t)) \right)
$$

$$
= \lim_{\epsilon \to 0^+} \frac{1}{\epsilon} \left( \left( \nabla \log p_t(\mathbf{z}(t)) - \nabla \log |\det \mathcal{J}_{\boldsymbol{\Phi}_\epsilon}(\mathbf{z}(t))| \right) \frac{\partial \boldsymbol{\Phi}_{-\epsilon}(\mathbf{z}(t+\epsilon))}{\partial \mathbf{z}(t+\epsilon)} - \nabla \log p_t(\mathbf{z}(t)) \right)
$$

$$
= \nabla \log p_t(\mathbf{z}(t)) \lim_{\epsilon \to 0^+} \frac{1}{\epsilon} \left( \left( \frac{\partial \boldsymbol{\Phi}_\epsilon(\mathbf{z}(t))}{\partial \mathbf{z}(t)} \right)^{-1} - \mathbf{I} \right) - \lim_{\epsilon \to 0^+} \frac{1}{\epsilon} \left( \nabla \log |\det \mathcal{J}_{\boldsymbol{\Phi}_\epsilon}(\mathbf{z}(t))| \left( \frac{\partial \boldsymbol{\Phi}_\epsilon(\mathbf{z}(t))}{\partial \mathbf{z}(t)} \right)^{-1} \right)
$$

$$
= - \nabla \log p_t(\mathbf{z}(t)) \frac{\partial \mathbf{f}(\mathbf{z}(t),t)}{\partial \mathbf{z}(t)} - \nabla \left( \boldsymbol{\nabla} \cdot \mathbf{f}(\mathbf{z}(t),t) \right),
$$

$$(16)$$

where the two limits are derived in detail:

$$
\lim_{\epsilon \to 0^+} \frac{1}{\epsilon} \left( \left( \frac{\partial \Phi_\epsilon(\mathbf{z}(t))}{\partial \mathbf{z}(t)} \right)^{-1} - \mathbf{I} \right)
$$

$$
= \lim_{\epsilon \to 0^+} \frac{1}{\epsilon} \left( \left( \frac{\partial}{\partial \mathbf{z}(t)} (\mathbf{z}(t) + \epsilon \mathbf{f}(\mathbf{z}(t),t) + o(\epsilon^2) + \dots) \right)^{-1} - \mathbf{I} \right)
$$

$$
= \lim_{\epsilon \to 0^+} \frac{1}{\epsilon} \left( \left( \mathbf{I} + \epsilon \frac{\partial \mathbf{f}(\mathbf{z}(t),t)}{\partial \mathbf{z}(t)} + o(\epsilon^2) + \dots \right)^{-1} - \mathbf{I} \right)
$$

$$
= \lim_{\epsilon \to 0^+} \frac{1}{\epsilon} \left( \left( \mathbf{I} - \epsilon \frac{\partial \mathbf{f}(\mathbf{z}(t),t)}{\partial \mathbf{z}(t)} + o(\epsilon^2) + \dots \right) - \mathbf{I} \right) \qquad \text{(inverse by geometric power series expansion)}
$$

$$
= \lim_{\epsilon \to 0^+} -\frac{\partial \mathbf{f}(\mathbf{z}(t),t)}{\partial \mathbf{z}(t)} + o(\epsilon) + \dots
$$

$$
= -\frac{\partial \mathbf{f}(\mathbf{z}(t),t)}{\partial \mathbf{z}(t)},
$$

and

$$
\lim_{\epsilon \to 0^+} \frac{1}{\epsilon} \left( \nabla \log |\det \mathcal{J}_{\boldsymbol{\Phi}_\epsilon}(\mathbf{z}(t))| \left( \frac{\partial \Phi_\epsilon(\mathbf{z}(t))}{\partial \mathbf{z}(t)} \right)^{-1} \right)
$$

$$
= \lim_{\epsilon \to 0^+} \frac{1}{\epsilon} \left( \nabla \log |\det \mathcal{J}_{\boldsymbol{\Phi}_\epsilon}(\mathbf{z}(t))| \left( \mathbf{I} - \epsilon \frac{\partial \mathbf{f}(\mathbf{z}(t),t)}{\partial \mathbf{z}(t)} + o(\epsilon^2) + \dots \right) \right)
$$

$$
= \lim_{\epsilon \to 0^+} \frac{\nabla \log |\det \mathcal{J}_{\boldsymbol{\Phi}_\epsilon}(\mathbf{z}(t))|}{\epsilon} - \underbrace{\lim_{\epsilon \to 0^+} \nabla \log |\det \mathcal{J}_{\boldsymbol{\Phi}_\epsilon}(\mathbf{z}(t))|}_{\nabla \mathbf{1} \to \mathbf{0}} \frac{\partial \mathbf{f}(\mathbf{z}(t),t)}{\partial \mathbf{z}(t)} \qquad (17)
$$

$$
= \nabla \lim_{\epsilon \to 0^+} \frac{\log |\det \mathcal{J}_{\boldsymbol{\Phi}_\epsilon}(\mathbf{z}(t))|}{\epsilon}
$$

$$
= \nabla \left( \boldsymbol{\nabla} \cdot \mathbf{f}(\mathbf{z}(t),t) \right).
$$

$\square$

Therefore, $\nabla \log p(\mathbf{x},t)$ follows a linear matrix differential equation, where the linear matrix is defined by the Jacobian $\frac{\partial \mathbf{f}(\mathbf{z}(t),t)}{\partial \mathbf{z}(t)}$ and the bias term is the gradient of divergence of the differential function $\nabla (\boldsymbol{\nabla} \cdot \mathbf{f}(\mathbf{z}(t),t))$.

### A.6 INTERPRETING ASCENT REGULARIZATION AS SCORE MATCHING OBJECTIVE

To show the ascent regularization in eq.(9) and eq.(10) relates to the score matching objective, we first assume a diffusion process defined via a stochastic differential equation (SDE):

$$
\mathrm{d}\mathbf{z}(t) = \mathbf{h}(\mathbf{z}(t),t) + g(t)\mathrm{d}\mathbf{W}(t), \mathbf{z}(0) = \mathbf{x}; \mathbf{x} \sim p(\mathbf{x}), \qquad (18)
$$

where $\mathbf{W}_t$ is Brownian motion and we denote $p_t(\mathbf{z}(t))$ as the marginal distribution at time $t$ and $\mathcal{P}_T$ as the path measure of the SDE up to time $T$.

(Anderson, 1982) shows the reverse time process is also a diffusion process which shares the same marginals as the forward process:

$$d\mathbf{z}(t) = \left(\mathbf{h}(\mathbf{z}(t), t) - g^2(t)\nabla \log p_t(\mathbf{z}(t))\right) dt + g(t)d\tilde{\mathbf{W}}(t), \mathbf{z}(T) \sim p_T, \tag{19}$$

where $\tilde{\mathbf{W}}(t)$ is a reverse-time Brownian motion. The reverse-time diffusion introduces the conditional path measure $\mathcal{P}(\cdot|\mathbf{z}(T))$. As the score function, $\nabla \log p_t(\mathbf{z}(t))$, is generally unknown for an arbitrary diffusion process, we approximate the reverse-time diffusion by a secondary reverse-time diffusion process by a parametric score function:

$$d\mathbf{z}(t) = \left(\mathbf{h}(\mathbf{z}(t), t) - g^2(t)s_{\boldsymbol{\theta}}(\mathbf{z}(t), t)\right) dt + g(t)d\tilde{\mathbf{W}}(t), \mathbf{z}(T) \sim p_T, \tag{20}$$

which induces the conditional path measure $\tilde{\mathcal{P}}_T^{\boldsymbol{\theta}}(\cdot|\mathbf{z}(T))$ to approximate $\mathcal{P}_T(\cdot|\mathbf{z}(T))$.

Under some regularity conditions that permit the definition of Radon-Nikodym derivative, $d\mathcal{P}_T(\cdot|\mathbf{z}(T))/d\tilde{\mathcal{P}}_T^{\boldsymbol{\theta}}(\cdot|\mathbf{z}(T))$, Girsanov theorem gives the expectation of KL divergence between two path measures:

$$\mathbb{E}_{p_T}\left[\mathrm{KL}(\mathcal{P}_T(\cdot|\mathbf{z}(T))\|\tilde{\mathcal{P}}_T^{\boldsymbol{\theta}}(\cdot|\mathbf{z}(T)))\right] = -\mathbb{E}_{\mathcal{P}}\left[\log \frac{d\tilde{\mathcal{P}}_T^{\boldsymbol{\theta}}(\cdot|\mathbf{z}(T))}{d\mathcal{P}_T(\cdot|\mathbf{z}(T))}\right]$$

$$=\mathbb{E}_{\mathcal{P}}\left[\int_0^T g(t)\left(s_{\boldsymbol{\theta}}(\mathbf{z}(t), t) - \nabla \log p_t(\mathbf{z}(t))\right) d\bar{\mathbf{W}}_t + \frac{1}{2}\int_0^T g^2(t)\|s_{\boldsymbol{\theta}}(\mathbf{z}(t), t) - \nabla \log p_t(\mathbf{z}(t))\|^2 dt\right]$$

$$=\frac{1}{2}\mathbb{E}_{\mathcal{P}}\left[g^2(t)\int_0^T \|s_{\boldsymbol{\theta}}(\mathbf{z}(t), t) - \nabla \log p_t(\mathbf{z}(t))\|^2 dt\right].$$

Using the chain rule of KL divergence, we can show the KL divergence between two path measures:

$$\mathrm{KL}(\mathcal{P}_T\|\tilde{\mathcal{P}}_T^{\boldsymbol{\theta}})$$

$$=\mathrm{KL}(p_T(\mathbf{z}(T))\|\mu(\mathbf{z}(T))) + \mathbb{E}_{p_T}\left[\mathrm{KL}(\mathcal{P}_T(\cdot|\mathbf{z}(T))\|\tilde{\mathcal{P}}_T^{\boldsymbol{\theta}}(\cdot|\mathbf{z}(T)))\right]$$

$$=\mathrm{KL}(p_T(\mathbf{z}(T))\|\mu(\mathbf{z}(T))) + \frac{1}{2}\mathbb{E}_{\mathcal{P}}\left[g^2(t)\int_0^T \|s_{\boldsymbol{\theta}}(\mathbf{z}(t), t) - \nabla \log p_t(\mathbf{z}(t))\|^2 dt\right] \tag{21}$$

$$=\mathrm{KL}(p(\mathbf{x})\|\tilde{p}(\mathbf{x})) + \frac{1}{2}\mathbb{E}_{\mathcal{P}}\left[\int_0^T g^2(t)\|s_{\boldsymbol{\theta}}(\mathbf{z}(t), t) - \nabla \log p_t(\mathbf{z}(t))\|^2 dt\right].$$

Assume that the parametric dynamics $\mathbf{f}(\mathbf{z}(t), t; \boldsymbol{\theta}) = \nabla \log \mu(\mathbf{z}(t)) - s_{\boldsymbol{\theta}}(\mathbf{z}(t), t)$ has the similar structure as the optimal dynamics in eq.(6) as $s_{\boldsymbol{\theta}}(\mathbf{z}(t), t)$ to approximate $\nabla \log p_t(\mathbf{z}(t))$ and $g(t) \equiv \sqrt{2\lambda}$, then we recover the total learning objective with ascent regularization coefficient $\lambda$ in eq.(9). Therefore, the total objective is equivalent to minimize the KL divergence of two path measures on the joint (infinite) variable space. Similar analysis can also be applied to the objective in eq.(10).

When $\lambda = \beta^{-1} = \frac{g^2(t)}{2}$ and learned score $s_{\boldsymbol{\theta}}(\mathbf{z}(t), t)$ matches to $\nabla \log p_t(\mathbf{z}(t))$ so that $\mathbf{h}(\mathbf{z}(t), t) = \nabla \log \mu(\mathbf{z}(t))$, then SDE in eq.(18) becomes the overdamped Langevin dynamics in eq.(13) as well as optimal ACNF (eq.(6)) with critical damping dynamics, i.e. $\lambda = \beta^{-1} = 1$.

As the ascent regularization can be interpreted as a score matching objective, it is possible to implement Algorithm 1 and Algorithm 2 in a more time efficient way for training like (Lu et al., 2022; Song et al., 2021). However, note that the explicit score matching objective can hardly be used directly in the implementation as $\nabla \log p_t(\mathbf{z}(t))$ is intractable in general and requires to be evaluated e.g. via score function integral in ascent regularization. (Lu et al., 2022; Song et al., 2021; 2020; Ho et al., 2020) use its surrogates, e.g. denoising score matching. To enable practical training, denoising score matching objective relies on the explicit form of conditional (noised) distributions $\nabla \log p_{t|0}(\mathbf{z}(t)|\mathbf{z}(0))$, e.g. Gaussian. For image or data generation tasks, Gaussian assumption may

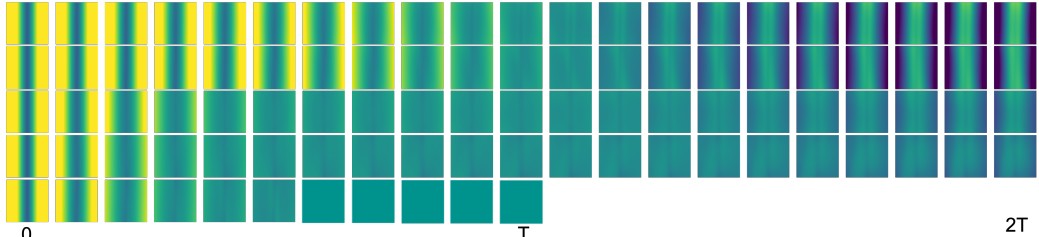

Figure 10: Comparison of log potential field, $\log V(\mathbf{z}(t), t)$, evaluated on trained vanilla CNF, RNODE with regularization coefficient as 0.1 and ACNF models with regularization coefficient $\lambda$ as 0.1 and 1 for 2-modal Gaussian mixture along flow at $t \in [0, 2T]$ and the numerical PDE solutions of eq.(7). Color indicates the value of field: turquoise is 0, and the lighter the color is the larger the value is, and vice versa.

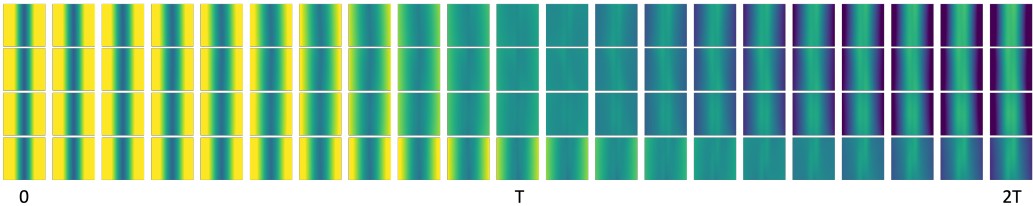

Figure 11: Comparison of log potential field, $\log V(\mathbf{z}(t), t)$, evaluated on trained vanilla CNF and RNODE (Finlay et al., 2020) models with $T = 1$ for 2-modal Gaussian mixture along the flows at $t \in [0, 2T]$ as Figure 4. The kinetic energy regularization coefficients are $0, 0.01, 0.1, 1$ respectively. Color indicates the value of field: turquoise is 0, and the lighter the color is the larger the value is, and vice versa.

not seem so limited as long as the chain of discrete transformation is adequately long for adequate expressivity of the marginal distribution at $T$. However, for inference tasks e.g. using flows as variational approximation or annealed sampler, constraining the distribution induced by flows with Gaussian assumption can hinder their approximate potential for true posterior.

### A.7 ANALYSIS ON A TOY EXAMPLE: FROM A GAUSSIAN TO A MIXTURE OF GAUSSIAN

Before we deploy ACNF for complex distributions, we first demonstrate its validity on a simpler problem: to learn a 2-modal Gaussian mixture with a standard Gaussian as the base distribution. Since the density of the target distribution is known in this case, we can numerically solve the potential field $V(\mathbf{z}, t)$ for $t \in [0, T]$ in eq.(7) even though the exact solution is still hard to obtain for this simple case.

The PDE solution presented in Figure 4 and Figure 10 is implemented using `py-pde` package. A fixed Cartesian grid is used which has the same center locations as the other potential fields evaluated by density estimations. The PDE solver in `py-pde` uses the finite difference method, and we choose explicit solver to keep simulation simple.

To define the parametric dynamics function for training, we use hypernetworks (Ha et al., 2016) that a smaller network generates the weights of layers. This architecture is suitable to demonstrate ACNFs as the function of dynamics is supposed to evolve with time via changing the weights by the hypernetworks. We follow the same implementation of hypernetworks as Neural ODE and use `torchdiff` for ODE solution and adjoint method. [1]

The last row of Figure 10 as Figure 4 shows the logarithm of the potential solutions, while the rest show the log potential field of learned flows evaluated by the ratio $p(\mathbf{x})/\tilde{p}_t(\mathbf{x})$ when training $T$ is set as 10.

---

[1] https://github.com/rtqichen/torchdiffeq

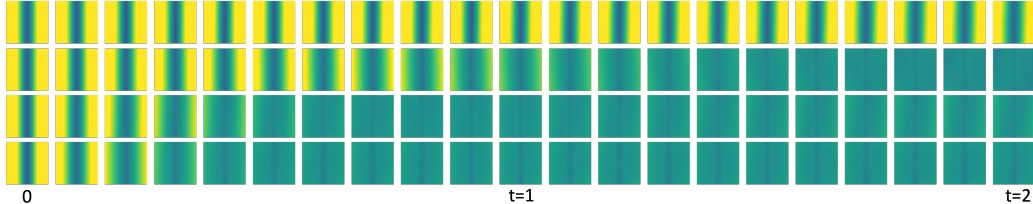

Figure 12: Comparison on log potential field, $\log V(\mathbf{z}(t), t)$ of trained vanilla CNF and ACNF models with $T = 5$ for 2-modal Gaussian mixture, evaluated along the flows at $t \in [0, 2]$ as Figure 4. The ascent regularization coefficients $\lambda$ are $0, 0.01, 0.1, 1$ respectively.

Without ascent regularization, the potential field converges slower and only reaches close to a uniform field at $T$. After $T$, some areas start to be under/over-represented when the learned flow continues to move samples towards the center of the field. Nevertheless, the flows learned with ascent regularization transform densities faster to the target distribution. When the ascent regularization coefficient $\lambda$ is 1, the evolution of the potential fields is very similar to that of PDE solutions which indicates the learned flow is close to the optimal ascent continuous normalizing flow.

Apart from vanilla CNF, we train RNODE models to demonstrate the effect of kinetic energy regularization on the transformation of distributions. As known from (Finlay et al., 2020), the optimal flow that minimizes $L_2$ transport cost induces straight sample trajectories and samples travel with constant speeds. Figure 11 shows the flows of RNODE models trained under the same configurations as Figure 4, and the kinetic energy regularization coefficients are $0, 0.01, 0.1, 1$ respectively.

Although RNODEs learn simpler ODE functions with lower NFEs compared to the flow without regularization, these flows do not induce the transformed distributions to converge faster. They are even slower at larger regularization coefficients. Like vanilla CNF, RNODE does not prevent the distribution to deteriorate after $T$. NFEs for each flow in Figure 11, at the time that the transformed distribution gives the maximum estimated log-likelihood, are $38, 38, 36, 32$, while the flows by ACNF are $26, 32, 36$ under $\lambda = 0.01, 0.1, 1$, nevertheless ascent regularization does not explicitly regularize for simpler ODE functions. We also tried Frobenius norm regularization on the Jacobian as suggested by (Finlay et al., 2020), HJB regularization (Onken et al., 2021; Yang and Karniadakis, 2020), second-order regularization (Kelly et al., 2020), however, the evolution of potential fields under these regularizations does not differ much to that of vanilla CNF and RNODEs as shown.

To demonstrate the effect of the length of flow $T$ in training configuration, we train vanilla CNF and ACNFs with other flow length, e.g. $T = 5$ and ascent regularization factors as $0, 0.01, 0.1, 1$, and evaluate the learned flows at $t \in [0, 2]$ as Figure 4. Under some suitable condition that there exists an optimal ACNF between the base and the target distributions, the flow is almost independent to the choice of flow length $T$. Comparing Figure 12 with $T = 5$ to Figure 4 with $T = 1$ but testing both on $t \in [0, 2]$, the flow by vanilla CNF is idle at early stage for $T = 5$ and is very sensitive to the choice of $T$, while the flows with ascent regularization are almost independent to the choice of $T$, which possibly makes tedious model selection on different $T$ or optimizing $T$ (Ghosh et al., 2020; Du et al., 2022) no longer necessary.

### A.8 DENSITY ESTIMATION ON 2D TOY DISTRIBUTIONS

Like Section A.7, we specify dynamics model by hypernetworks and all hypernetworks are defined by one hidden layer with 32 units and 64 for the width of hypernetworks to learn all 2-dimensional distributions.

As shown in the last section, the flows learned with ascent regularization are almost insensitive to $T$ for Gaussian mixture. To examine whether this conclusion still applies to more complex distributions, we retrain ACNF models with ascent regularization coefficients $\lambda = 0.0001, 0.0005, 0.001, 0.005$ under different flow lengths $T = 10, 5, 1, 0.5$. Figure 13 ($T = 5$) and Figure 14 ($T = 1$) shows the evolution of the density estimations for each model at $t \in [0, 2T]$ like Figure 5 ($T = 10$). When decreasing $T$ from 10 to 5, the density estimations are almost identical under the same regularization coefficients. When $T$ decreases from 5 to 1, the highlighted area shrinks slightly at low regularization coefficients, e.g. $0.0001, 0.0005$. Model trained with a smaller $T$ may require a larger $\lambda$ to have

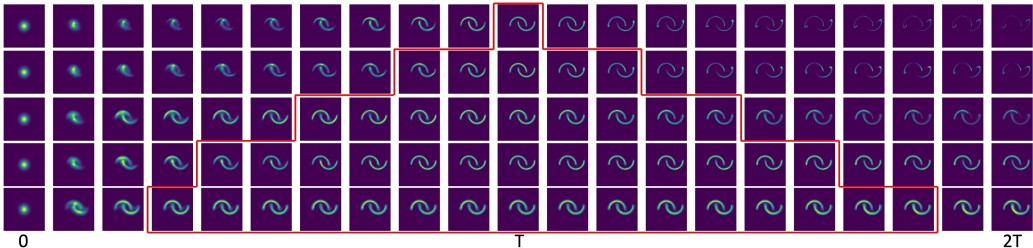

Figure 13: Comparison on density estimations of trained vanilla CNF and ACNFs with regularization coefficients $\lambda = 0.0001, 0.0005, 0.001, 0.005$ and $T = 5$ on 2-moon distribution at $t \in [0, 2T]$.

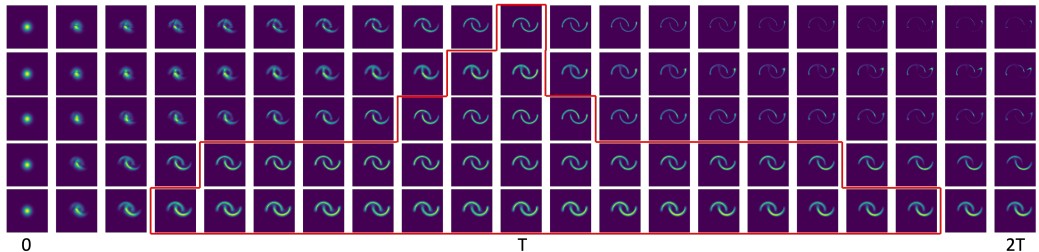

Figure 14: Comparison on density estimations of trained vanilla CNF and ACNFs with regularization coefficients $\lambda = 0.0001, 0.0005, 0.001, 0.005$ and $T = 1$, on 2-moon distribution at $t \in [0, 2T]$.

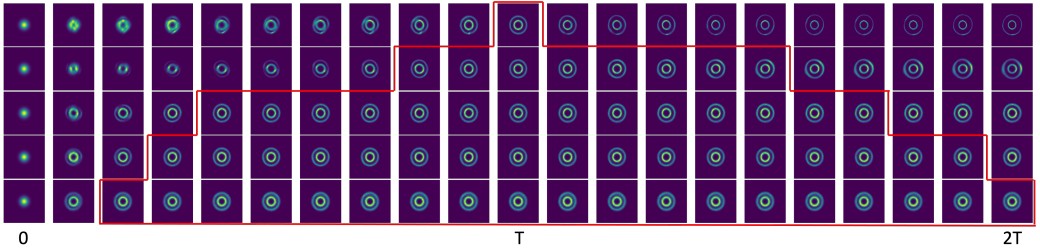

Figure 15: Comparison on density estimations of trained vanilla CNF and ACNFs with regularization coefficients $\lambda = 0.0001, 0.0005, 0.001, 0.005$ and $T = 10$ on 2-circle distribution at $t \in [0, 2T]$.

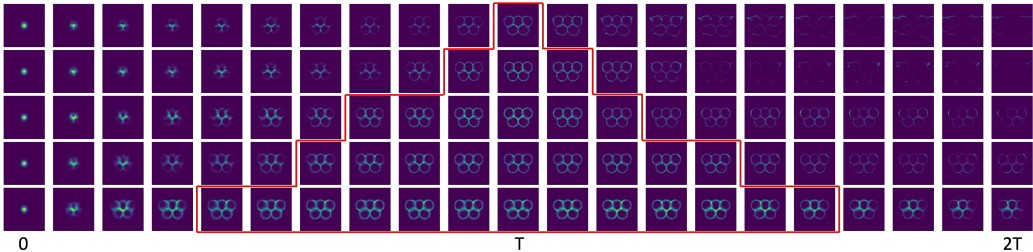

Figure 16: Comparison on density estimations of trained vanilla CNF and ACNFs with regularization coefficients $\lambda = 0.0001, 0.0005, 0.001, 0.005$ and $T = 10$ on Olympics distribution at $t \in [0, 2T]$.

similar regularization that with a larger $T$ as the flow length serves as an implicit regularization factor. Although the effect of regularization depends slightly more on the choice of $T$ for complicated distributions, the flows by ACNF are still much less sensitive to $T$, compared to that by CNF.

Apart from the 2-moon distribution, we show the density estimations of learned vanilla CNF and ACNF with different regularization coefficients for modeling 2-circle, Olympics and checkerboard distributions in Figure 15, Figure 16 and Figure 17. They show that ascent regularization is effective in learning different distributions that a larger coefficient induces densities to converge faster to the target distributions and prevents them from deterioration. Comparing across different distributions,

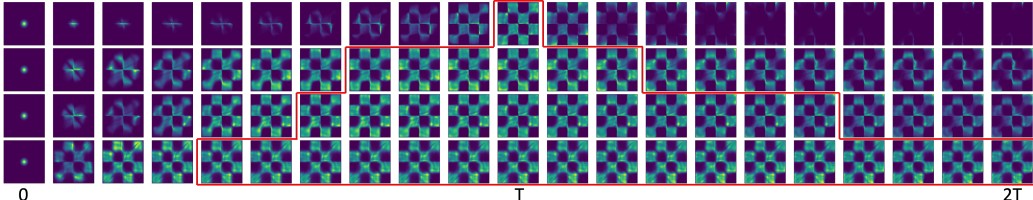

0                                                                    T                                                                    2T

Figure 17: Comparison on density estimations of trained vanilla CNF and ACNFs with regularization coefficients $\lambda = 0.0001, 0.0005, 0.001$ and $T = 10$ on checkerboard distribution at $t \in [0, 2T]$.

| Dataset | # hypernetworks layers | encoding dim | $T$ | # flow steps | batch size |
|---|---|---|---|---|---|
| POWER | 2 | 6 | 1 | 5 | 10000 |
| GAS | 3 | 4 | 5 | 5 | 1000 |
| HEPMASS | 3 | 10 | 1 | 10 | 10000 |
| MINIBOONE | 4 | 10 | 1 | 1 | 1000 |
| BSDS300 | 4 | 10 | 5 | 2 | 10000 |

Table 3: Model architectures of ACNFs for density estimations on tabular data reported in Table 1.

the highlighted areas are larger for 2-moon, 2-circle and checkerboard distribution than Olympics distributions, since the Olympics distribution is more challenging and requires a relatively large regularization coefficient.

## A.9    DENSITY ESTIMATION ON TABULAR DATASETS

For tabular datasets, we follow the experiment setup and model configurations as recommended by FFJORD (Grathwohl et al., 2018) and all data are pre-processed according to (Papamakarios et al., 2017). We found that the concatenate layer used in FFJORD, that concatenates time $t$ and states $\mathbf{z}(t)$ as a flat input vector for differential function, dilutes the ascent regularization on the parameters, especially when data dimensions are high, e.g. for MINOBOONE and BSDS300 datasets. Nevertheless, the hypernetwork architecture used in previous sections, even a deeper one, turns out to be inadequate to reach a similar log-likelihood evaluation as FFJORD and slow to train. To tackle this issue, we use an encoder to encode states $\mathbf{z}(t)$ to a lower dimension and apply the weights by the hypernetworks on the encodings and later a decoder maps the transformed encodings back to the data dimension. We summary model architectures and training configurations for each dataset in Table 3.

## A.10    ACNFs AS ANNEALED SAMPLER FOR UNBIASED SAMPLING AND ESTIMATE OF NORMALIZATION CONSTANT

To extend ACNF annealed sampler with stochasticity, we replace the discrete NF blocks in SNF by the discrete realization of each adaptive step of ACNF and each is followed with a stochastic block by e.g. discrete Langevin flow or MCMC flow as in SNF. The original importance weight update for discrete flows also needs to be replaced by the integral of negative divergence of dynamics and resampling steps are added as AFT (Arbel et al., 2021). The complete algorithm is summarized in Algorithm 3.

Figure 18 shows the generated samples of all different methods as reported in Figure 8 plus adding MC steps on top of trained ACNF to form SNF models by Algorithm 3. Like quantitative evaluation shown in Figure 8, learned ACNFs with regularization coefficient $\lambda = 0.01$ has distinctly faster convergence than CNF, best tuned linear annealed target and less regularized ACNFs, but uses less computation. The add-on MC steps on trained ACNF boasts the convergence slightly as shown by the last two rows. Although diffeomorphism constraint does not show much effect on limiting the expressiveness of CNF/ACNF in this experiment, adding stochastic blocks is still very beneficial especially at the beginning stage of the flows.

---

**Algorithm 3** Asymptotically unbiased sampler with learned ACNF $\mathbf{f}_{\boldsymbol{\theta}}$

---

**Require:** parameteric dynamics of ACNF generation flow $\mathbf{f}_{\boldsymbol{\theta}}$, base distribution $\mu$, target distribution up to the normalization constant $\pi(\cdot) = \gamma(\cdot)/Z$, length of flow $T$, number of samples $N$, MC step size $\epsilon$, number of MC steps $J$

sample N samples from base distribution $\{\mathbf{z}_0^i\}_{i=1:N} \sim \mu = q_0$

set $\log w_0^i = -\log \mu(\mathbf{z}_0^i)$, $t_0 = 0$

**while** $t_k < T$ **do**

    ODE solver chooses step size $\Delta t_k$, if $t_k = t_{k-1} + \Delta t_k < T$ else $t_k = T$

    Integrate augmented states $[\mathbf{z}^i(t), \log q_t(\mathbf{z}^i(t))]$ using generation dynamics $\mathbf{f}_{\boldsymbol{\theta}}$ until $t_k$ from the initial $[\mathbf{z}_{k-1}^i, \log q_t(\mathbf{z}_{k-1}^i)]$ at $t_{k-1}$

    $\Delta S_{k,f}^i = \log q_{t_{k-1}}(\mathbf{z}_{k-1}^i) - \log q_{t_k}(\mathbf{z}^i(t_k))$

    $\mathbf{z}_k^i = \mathbf{z}^i(t_k)$

    MCMC update with $\pi$ invariant kernel via Metropolis-Hastings:

    **for** $j = 1, \ldots, J$ **do**

        propose $\mathbf{z'}_k^i = \mathbf{z}_k^i + \epsilon \eta^i, \eta^i \sim \mathcal{N}(\mathbf{0}, \mathbf{I}), \forall i$

        $a^i = \gamma(\mathbf{z'}_k^i)/\gamma(\mathbf{z}_k^i), \forall i$

        **if** $\xi^i < a^i$, $\xi^i \sim U(0, 1)$ **then**

            update $\mathbf{z}_k^i = \mathbf{z'}_k^i$

        **end if**

    **end for**

    $\Delta S_{k,s}^i = \log \gamma(\mathbf{z}^i(t_k)) - \log \gamma(\mathbf{z}_k^i), \forall i$

    Update weights $\log w_k^i = \log w_{k-1}^i + \Delta S_{k,f}^i + \Delta S_{k,s}^i$

    Resample $\mathbf{z}_k^i$ according to normalized weights $\tilde{w}_k^i = w_k^i/(\sum_i w_k^i)$

    Update weights $w_k^i = 1/N$

**end while**

---

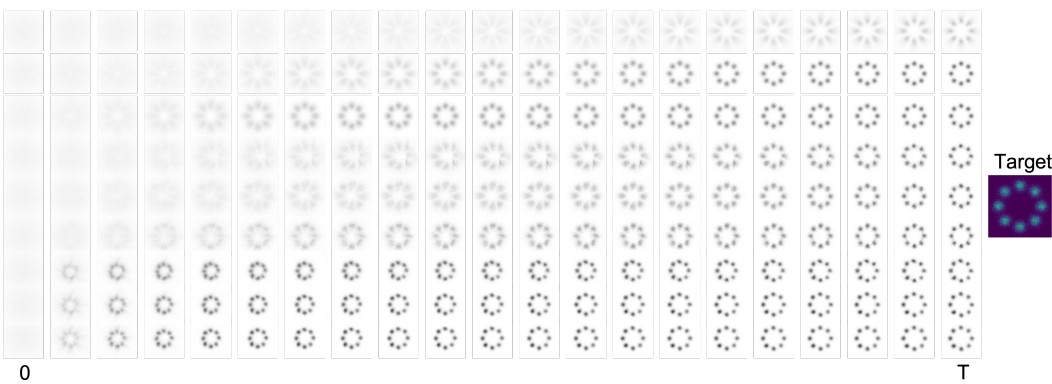

Figure 18: Comparisons of generated samples from different methods on 2D Gaussian mixture distribution with 8 components as Figure 8. From top to bottom, samples are from: (1-3) linear annealing importance sampler with $\{170, 25, 10\}$ MC steps between each annealing target; (4) CNF; (5-7) ACNF with ascent regularization factor $\lambda = 0.0001, 0.001, 0.01$, (8-9) SNF with trained ACNF $\lambda = 0.01$ (as 7th row) and $\{1, 5\}$ MC step as the stochastic block as Algorithm 3.

## A.11   VARIATIONAL INFERENCE WITH ACNFS

Our experiment setup mimics (Grathwohl et al., 2018), and the encoder and decoder are defined by 7-layer neural networks with specified latent dimension as $64$. The first 6 layers of the encoder are implemented as gated convolutional networks and the last one is a linear layer to output mean and diagonal covariance. For the decoder, the first 6 layers are also gated convolutional networks while the last layer is a vanilla convolutional network. We define the length of flow for both VAE-FFJORD and VAE-ACNF as $T = 1$ and the number of steps as 2. The networks for modeling differential function of flows are the modified hypernetworks as for the tabular datasets, with 4 layers, and the

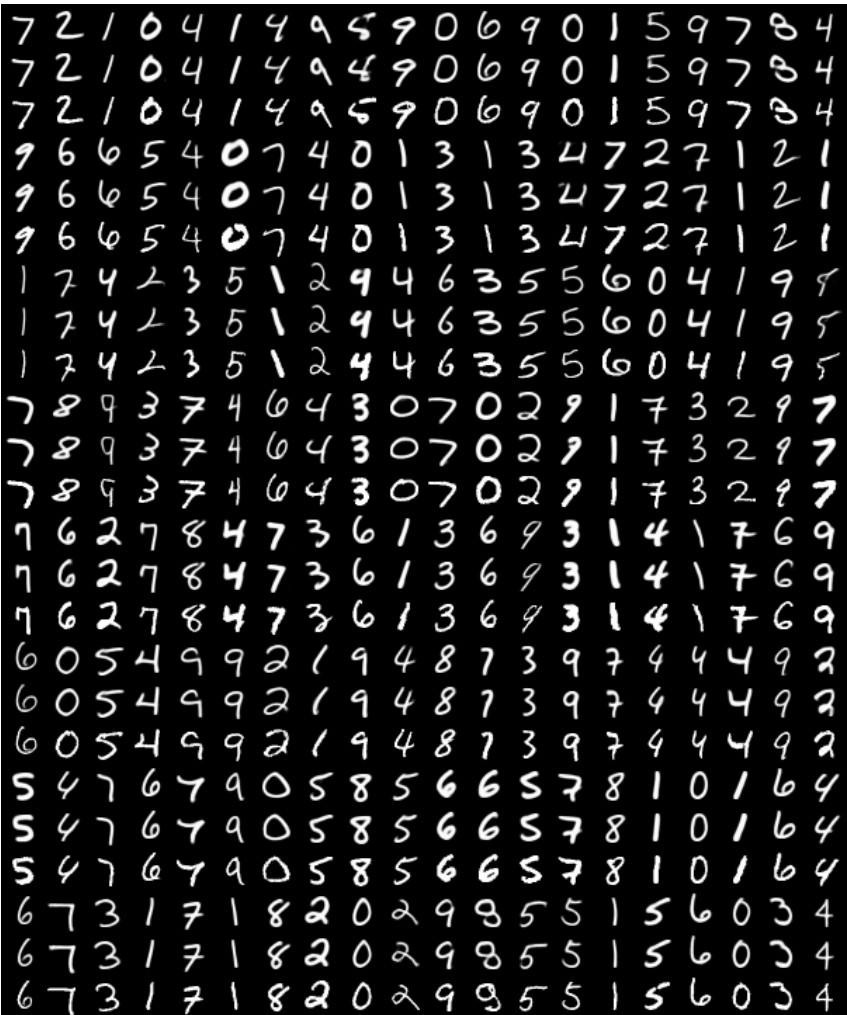

Figure 19: More reconstructed samples from VAE-ACNF, vanilla VAE and original data. The first row of three is the reconstruction from VAE-ACNF, the second one is the reconstruction from vanilla VAE while the last one is the original data samples.

activation function is tanh. All models reported in Table 2 are trained under the same learning rate as 0.001 , `Adam` optimizer and batch size as 100.

Figure 19 shows more reconstructed samples from VAE-ACNF and vanilla VAE, with comparison of original data. In general, the reconstructions from VAE-ACNF are smoother than the ones from vanilla VAE and original data samples. Figure 9 shows some challenging examples for VAE to reconstruct. VAE-ACNF tends to reconstruct images by adding more details, not only to make it smoother, but also to possibly strengthen their identity of classes. Furthermore, due to the coarse variational approximation, some reconstructions of VAE fail to retain their features in original data and change the identity of classes.

