# OpenReview forum: "Learning Continuous Normalizing Flows For Faster Convergence To Target Distribution via Ascent Regularizations"
_ICLR.cc/2023/Conference — ICLR 2023 poster_

### Official Review · Reviewer_niTY · 2022-10-25

**Confidence:** 4
**Correctness:** 3
**Technical Novelty And Significance:** 2
**Empirical Novelty And Significance:** 3
**Recommendation:** 6

**Clarity, Quality, Novelty And Reproducibility:**

Most part of the paper is well-written with hight quality. The idea from this paper is too totally new, for instance similar ideas are in Stein Variational Gradient Descent method or other variational method. No problem in the reproducibility part.

**Strength And Weaknesses:**

Pros:The paper proposed ACNFs, a new class of CNFs, that define flows with monotonic convergence toward a target distribution. The authors derive the dynamics for the steepest ACNF and propose a practical implementation to learn parameteric ACNFs via ascent regularization. The learned ACNFs illustrate three beneficial behaviors: 1) faster convergence to the target distribution with less computation; 2) self-stabilization to mitigate performance deterioration; 3) insensitivity to flow training length T . Experiments on both toy distributions and real-world datasets demonstrate the effectiveness of ascent regularization on learning ACNFs for various purposes.

Cons: The definition of \tilde p_t, at least in the main body of the paper, say in page 3, is not very clear. I would suggest the authors to make the derivation part before more clear, say those around prop. 1.

**Summary Of The Paper:**

The authors  propose a new class of continuous Normalizing flows (NFs), ascent continuous normalizing flows (ACNFs), that makes a base distribution converge faster to a target distribution.  The learned ACNFs demonstrate faster convergence towards the target distributions, therefore, achieving better density estimations, unbiased sampling and variational approximation at lower computational cost.

**Summary Of The Review:**

This paper is above the average. I would suggest to accept this paper.

---

> ### Author Response · Authors · 2022-11-09
> **Thanks for your insightful comments and we would like to address your concerns and questions.**
>
> We thank the reviewer for providing valuable feedback to improve the work. We will address concerns and questions from the reviewer.
>
> 1. Clarification on the definition and motivation on time-$t$ estimated likelihood, $\tilde{p}(\mathbf{x})$
>
> $\tilde{p}_t$ is the t-length flow approximate of data distribution $p_0(x)$. When flow length is 0, it is the base distribution, i.e. $\tilde{p}_0 = \mu$. When $t$ increases, as $p_t$ gets closer towards base distribution $\mu$, time-$t$ estimate $\tilde{p}$ gets closer to data distribution p(x).
>
> Since most previous works focus on evaluating time-$T$ flow estimate only and no flow with arbitrary length is examined, the approximation error of $\log p_0(x)$ by integral (the equation between eq.(3) and eq.(4)) has been overlooked. As we need to demonstrate the performance of flow with length of any t, e.g. in Figure 1 and 2, defining $\tilde{p}_t$ allows us to avoid this ambiguity and evaluate at arbitrary $t$ without unnecessary computation.  We update Section 2 especially around definition of time-t estimated likelihood and flow illustration in Figure 3 in the revised version of submission.
>
> 2. Relation to SVGD?
>
> The gradient flow in our theorem 1 is under 2-Wasserstein metric while SVGD is defined in RKHS (reproducing kernel Hilbert space). Additionally, from a practical use perspective, the proposed ACNF is a model via learnable dynamics for e.g. exact inference algorithm in Section 6.1-6.2, e.g. annealed importance sampler in Section 6.3 and e.g. (parametric) variational inference algorithm in Section 6.4, while SVGD uses the Vlasov process to evolve samples from base distribution towards target distribution and can be viewed as a non-parametric variational inference algorithm.

---

### Official Review · Reviewer_XKEH · 2022-10-26

**Confidence:** 3
**Correctness:** 4
**Technical Novelty And Significance:** 3
**Empirical Novelty And Significance:** 3
**Recommendation:** 8

**Clarity, Quality, Novelty And Reproducibility:**

- The paper is written well and the ideas seem novel.
- The authors share some code but no unit tests on how to run yet. Hopefully, this will be fixed in the final submission to allow for reproducibility.


**Strength And Weaknesses:**

Strengths:

- The authors have formulated a novel way of regularizing continuous flows that not only makes the flow stable but also improves efficiency (in terms of NFEs) and final performance (in terms of NLL)
- The authors have backed their regularization functional with ample explanations and theoretical justification and followed it up with experiments that reflect the contributions they claim.

Weaknesses:
- While not specific to this work but continuous flows papers in general should discuss an important limitation: scaling to high-dimensional datasets like ImageNet or FFHQ. The authors show some results on MNIST but I think the stronger case should be made on high-dimensional images. If not, commenting on a possible roadmap to get there might also be useful.
- I am unsure how the computational cost compares wrt to previous work. From Algorithms 1 and 2 it seems like integrating augmented states for z_t and then for log(p_t(z_t)) should incur a higher computational cost. It would be better if the authors could discuss the overall compute cost as well and not just NFE.
- Another thing to discuss is more theoretical: why should there exist a map from the data distribution to a base distribution that has the ascent continuous property? The topology of the support of the data distribution may not be homeomorphic to a ball in high-dimensional space (in case of Gaussian) ? Moreover even if it is homeomorphic, it is not obvious to me why should an ascent-continuous map always exist.


**Summary Of The Paper:**

The paper proposes a way to regularize continuous normalizing flows by regularizing them to have monotonically decreasing KL divergence wrt to the base distribution along the integration path. In doing so the paper builds on previous work of NODE, RNODE and CNFs and adds to the collection of papers around stabilizing continuous normalizing flows. Through a bunch of experiments on tabular datasets and some experiments on MNIST images the paper shows that their method can indeed lead to flows that converge faster (less NFEs) and are more stable.


**Summary Of The Review:**

I think the paper makes a good contribution to the bijective normalizing flows literature. It supports the claimed contribution with theory and appropriate empirical analysis.

---

> ### Author Response · Authors · 2022-11-09
> **Thanks for your insightful comments and we would like to address your concerns and questions.**
>
> We thank the reviewer for providing valuable feedback to improve the work. We will address concerns and questions from the reviewer, and answer them individually.
>
> 1. Discuss the limitation on continuous normalizing flows and the roadmap for scaling up to high-dimensional datasets like ImageNet or FFHQ:
>
> As the reviewer pointed out, CNF faces some difficulties in general to scale up for high-dimensional data due to the diffeomorphism nature and computation complexity for training. As ascent behavior has not been discussed widely in the community, we feels ACNFs are easier to explain and demonstrate as CNFs have more tools for rigid analysis.
>
> Although ACNF presented in this manuscript is proposed under CNFs, the concept of enforcing ascent behavior can be extended to diffusion models or score-based models or other stochastic flows, especially Wasserstein gradient flows, which have a very strong connection to SDEs. In Section A.6 in Appendix of the updated manuscript, we reinterpret the ascent regularization as a score matching objective to build a connection to score-based models. The added analysis can be the stepstone for introducing ascent regularization in the score-based models which potentially resolve the expressivity limit in CNFs. And it shows the possibility to implement Algorithm 1 and Algorithm 2 for ACNFs in more time efficient ways like [1].
>
> 2. Computation cost of ACNFs at inference and training phases?
>
> For inference time, the algorithm complexity and memory requirement of ACNF are the same as NODE, RNODE and other OT-Flows as only the forward integration in Algorithm 1 needs to compute z and log-likelihood simultaneously and the total computation cost is linear to NFEs.
>
> For training time by Algorithm 1 and Algorithm 2, however, the computational complexity is higher than other baselines due to the Jacobian estimate and the gradient of divergence in eq. (9 ) for integrating score of logp_z(z_t) in ascent regularization as discussed in the limit section. Besides, we find that in practice the wall time of training on ACNFs is not much higher than that of RNODE to reach similar model performance. The possible explanations are 1) The ascent regularization adjusts model distribution for the rapid convergence; 2) less number of function evaluations for integration.
>
>
> 3. Why should there exist a map from the data distribution to a base distribution that has the ascent continuous property?
>
> As ACNF is a special instance of CNF which is essentially a diffeomorphism, it must preserve topological properties and some manifolds like spheres are not homeomorphic to R^D. For the case that there exists a map from the data distribution to a base distribution, we can explain in short: from Brenier’s theorem, under a measure-preserving transformation, there exists a $\mu$ unique map $\nabla \phi^*$ pushing the starting distribution $\mu$ forward to $\nu$ satisfying $\nu = \nabla \phi^*$#$\mu$ and $\nabla \phi^*$ is the unique minimizer of Wasserstein-2 metric. We refer [2] for a more detailed explanation of the existence and uniqueness of the map.
>
>
> [1] Song, Yang, et al. "Maximum likelihood training of score-based diffusion models." Advances in Neural Information Processing Systems 34 (2021): 1415-1428.
>
> [2] Filippo Santambrogio. Optimal transport for applied mathematicians. Birkäuser, NY, 55(58- 63):94, 2015.

---

### Official Review · Reviewer_XQ2y · 2022-10-26

**Confidence:** 3
**Correctness:** 3
**Technical Novelty And Significance:** 3
**Empirical Novelty And Significance:** 3
**Recommendation:** 6

**Clarity, Quality, Novelty And Reproducibility:**

Novelty:
  - Combining a score function estimator with the Wasserstein gradient flow for regularizing CNFs seems novel and works well in practice.

Quality:
  - ACNF compares favorably to CNF and RNODE (an OT-based regularization for CNFs) on a variety of settings.
  - Being able to get close to the T=1 distribution with a shorter time interval is interesting.

Clarity:
  - Eq (4), the "estimated log-likelihood" warrants more clarification. From what I understand, it is an approximation to p_0(x) where instead of integrating all the way from 0 to T, this integrates only up to t and directly assumes the base distribution at time t. So it is equal to p_0(x) when t=T, but otherwise is not really related to p_t. The sentence before Eq (4) makes it seem like \tilde{p}_t is approximating p_t, but this isn't the case right?
  - It also isn't clear why \tilde{p}_t is being introduced. The sentence before Eq (4) states this is to avoid extra integration steps, but both Algorithms 1 and 2 seem to integrate all the way from 0 to T regardless. And the \tilde{p}_T in the training objective can be replaced by p_0. Can the authors clarify the importance of introducing \tilde{p}_t ?
  - Eq (7) seems out of place. I did not understand how this equation contributes to the paper but this takes up half of the section on ACNF. It might be better to have a dedicated Related Works section to discuss connections to other works. On the contrary, Eq (8) is interesting and should be emphasized more strongly I feel.
  - How are the NFEs computed? Is the ODE being solved with an adaptive solver?
  - Why is the model regularized for t > T? I did not understand why this is the case, since the model does not regularize beyond [0, T]. Also, is regularizing t > T useful for anything?

**Strength And Weaknesses:**

Strengths:
  - The proposed approach is interesting, as a way to mimic Wasserstein gradient flow paths. Using the time evolution of the score function is interesting as well.
  - The algorithm works well at regularizing the path and seems to help accelerate convergence towards the target distribution. This is shown across both MLE and VI settings, as well as in a toy setting where it is shown to improve upon MCMC in terms of the number of function evaluations.

Weaknesses:
  - I found the writing poor and can be significantly improved. Specifically, there seems to be multiple concepts being introduced \tilde{p}_t (Eq 4) and V (Eq 7) that do not appear in the experiments. See clarity section below for more comments. If I am mistaken, please correct me.

**Summary Of The Paper:**

This paper discusses ascent regularization for training continuous normalizing flows (CNFs). This is motivated from Wasserstein gradient flows and results in an interesting regularization that encourages the learned model to be similar to the target distribution around a large interval of time values. I find this to be an interesting alternative to optimal transport-based approach for reducing the compute cost to transport samples between a base distribution and a target distribution. Experiments include maximum likelihood on tabular data and as a variational inference model within a VAE.

**Summary Of The Review:**

I think this paper is interesting and has potential. However, I feel hesitant to recommend acceptance in its current form, particularly since multiple concepts and equations are introduced but I did not understand how these connect with the final training algorithm. My understanding is that the main contribution seems to be using the time evolution of the score function for regularizing the ODE to model the steepest descent direction on the KL, but the connection to \tilde{p}_t is unclear to me. If the authors can clarify this, I would be willing to change my rating.

---

> ### Author Response · Authors · 2022-11-09
> **Thanks for your insightful comments and we would like to address your concerns and questions.**
>
> We thank the reviewer for providing valuable feedback to improve the work. We will address concerns and questions from the reviewer, and answer them individually.
>
> 1. Clarification on the definition and motivation on time-$t$ estimated likelihood, $\tilde{p}(\mathbf{x})$
>
> $\tilde{p}_t$ is the t-length flow approximate of data distribution $p_0(x)$. When flow length is 0, it is the base distribution, i.e. $\tilde{p}_0 = \mu$. When $t$ increases, as $p_t$ gets closer towards base distribution $\mu$, time-$t$ estimate $\tilde{p}$ gets closer to data distribution p(x).
>
> Since most previous works focus on evaluating time-$T$ flow estimate only and no flow with arbitrary length is examined, the approximation error of $\log p_0(x)$ by integral (the equation between eq.(3) and eq.(4)) has been overlooked. As we need to demonstrate the performance of flow with length of any t, e.g. in Figure 1 and 2, defining $\tilde{p}_t$ allows us to avoid this ambiguity and evaluate at arbitrary $t$ without unnecessary computation.  We update Section 2 especially around definition of time-t estimated likelihood and flow illustration in Figure 3 in the revised version of submission.
>
> Although the main purpose of proposing $\tilde{p}_t$ is for evaluating/inference and the current implementation in Algorithm 1 integrates $\tilde{p}_t$ until $T$, sampling flow length $t$ in each iteration of training can be implemented like [1, 2] to decrease training time and $\tilde{p}_t$ can be used for training as well in this case.
>
> 2. Purpose of introduce eq.(7) (Wasserstein gradient flow PDE on potential ratio)
>
> eq.(7) serves as the basis or equivalent problem in the previous implementation for solving Wasserstein gradient flows. Previous works [3,4] mainly develop the solution based on JKO integration, however, it is not so easy to apply them for modeling data. Instead, ascent regularization provides an alternative to resolve the difficulties and make training ACNF more accessible. In Figure 4, we compare learned ACNF to the numerical solution of the PDE eq.(7) to prove that the ascent regularization is sufficient for learning close-to-optimal ACNF.
>
> 3. How NFEs are evaluated? Is adaptive solver used for ODE integration?
>
> We evaluate NFEs by counting how many times the dynamics function f is called during integral which also includes the error evaluation for choosing step size by the adaptive solver. The results reported in the manuscript are all run with an ode45 solver implemented in torchdiffeq package.
>
> 4. Why is the learned model regularized for t>T, as it is not regularized beyond [0,T] during training? How is the regularization on t>T useful in practice?
>
> As mentioned in Section 3 (the last indication of steepest dynamics of ACNF), the flow gradually slows down when p_t becomes closer to $\mu$ and it completely stops when $p_t = \mu$ everywhere. Therefore, ACNF regularizes itself for the distribution at $t>T$ even though it is trained only with explicit regularization at [0,T]. This bonus property has twofold benefits in practice: 1) As Figure 2 shows, time-t estimated log-likelihood is less sensitive to flow length around $T$ compared to vanilla CNFs, therefore, it is less prone to numerical error from the solvers; 2) This property is closely related to ergodicity in Markov Chain Monte Carlo (MCMC) or Bolzmann sampling that with infinitely long sampling chain (flow length here), distribution will converge to the invariant distribution (target distribution here) irrespective of the choice of initial distribution, which we demonstrate ACNF can actually capture this behavior partially.
>
>
> [1] Huang, Chin-Wei, et al. "Improving explorability in variational inference with annealed variational objectives." Advances in Neural Information Processing Systems 31 (2018).
>
> [2] Song, Yang, et al. "Maximum likelihood training of score-based diffusion models." Advances in Neural Information Processing Systems 34 (2021): 1415-1428.
>
> [3] Petr Mokrov, Alexander Korotin, Lingxiao Li, Aude Genevay, Justin M Solomon, and Evgeny Burnaev. Large-scale wasserstein gradient flows. Advances in Neural Information Processing Systems, 34, 2021.
>
> [4] Jiaojiao Fan, Amirhossein Taghvaei, and Yongxin Chen. Variational wasserstein gradient flow. Proceedings of the 39 th International Conference on Machine Learning, Baltimore, Maryland, USA, PMLR 162, 2022.

---

> > ### Comment · Reviewer_XQ2y · 2022-12-01
> > **Thanks for the response**
> >
> > Thank you for your response. The revised manuscript has become clearer and I have revised my score.
> >
> > Another thing that warrants some clarification: following the steepest descent direction only ensures p_t converges to \mu at t=infinity right? But the experiments are only for finite T, which may end up with very different signals from the maximum likelihood objective and the regularization (i.e. they cannot both be satisfied at the same time). Is my reasoning correct?
> >
> > What value of T is used in the experiments, and how did you know this was sufficient for convergence such that p_T is sufficiently close to mu?

---

> > > ### Author Response · Authors · 2022-12-06
> > > **Thanks for value questions and we would like to address your questions as following.**
> > >
> > > 1. Theoretically, for the optimal continuous flow defined by eq.(6), there exists a non-zero bias for a finite time $T$ flow and it becomes zero at $t \rightarrow \infty $. The bias converges at the speed that can be measured by the expected absolute difference between scores of $p_t$ and $\mu$. In practice, the discrete-time approximation of the continuous solution is needed for implementation and the optimal discrete-time transport plan exists for general Wasserstein gradient flows and can be solved via e.g. JKO scheme [1], which guarantees that the discrete-time solution does not necessarily rely on the infinite $T$ to achieve the unbiasness. We will add more explanation and discussion in the final version.
> > >
> > > 2. As the ascent regularization coefficient makes the trade-off between the maximum likelihood objective and ascent regularization, the two objectives may have different optima for a limited family of dynamics function. In the experiments, we have observed that in some cases, too large regularizations do impede maximum likelihood learning for finite $T$-flows, and an intermediate regularization gives a better balance between the faster convergence behavior and maximum likelihood. However, it does not mean that two objectives conflict with each other for a finite $T$ especially when the dynamics function is adequately flexible, as the ascent regularization is equivalent to maximize the likelihood $\tilde{p}_t$ at any $t < T$ while the maximum likelihood objective focuses on the likelihood $\tilde{p}_T$ at $T$.
> > >
> > > 3. We reported different $T$s in both toy (T=0.5, 1, 5, 10) and real dataset experiments (different $T$s for different datasets in Table 3 for fair comparison to SOTA). In theory, a larger $T$ makes $p_T$ more expressive, thus it can be closer to $\mu$. In our experiments especially in Section 6.1 and 6.2 and in previous work [2,3], as a rule of thumb, larger $T$s are usually beneficial for expressivity but disadvantageous for convergence. The ascent regularization effectively prevents the slow convergence due to large $T$ while still achieving the maximum likelihood learning.
> > >
> > > 4. Regarding the sufficiency of flow length (or expressivity, to be more precise) for general continuous-time flows, we also need to consider other factors like the flexibility of function family of dynamics $f$, step size of solvers, and the sufficient flow length $T$ may vary from dataset to dataset (e.g. different $T$ is used for different datasets in Section 6.2), thus it is difficult to give a concrete indication that what is a sufficient $T$ for general continuous normalizing flows and we refer [4] for some initial study on this topic.
> > >
> > > [1] Petr Mokrov, Alexander Korotin, Lingxiao Li, Aude Genevay, Justin M Solomon, and Evgeny Burnaev. Large-scale wasserstein gradient flows. Advances in Neural Information Processing Systems, 34, 2021.
> > >
> > > [2] Will Grathwohl, Ricky TQ Chen, Jesse Bettencourt, Ilya Sutskever, and David Duvenaud. Ffjord: Free-form continuous dynamics for scalable reversible generative models. In International Conference on Learning Representations, 2018.
> > >
> > > [3] Chris Finlay, Jörn-Henrik Jacobsen, Levon Nurbekyan, and Adam M Oberman. How to train your neural ode: the world of jacobian and kinetic regularization. In International Conference on Machine Learning, 2020.
> > >
> > > [4] Verine, Alexandre, et al. "On the expressivity of bi-Lipschitz normalizing flows." arXiv preprint arXiv:2107.07232 (2021).

---

### Public Comment · ~Kaiwen_Zheng2 · 2022-11-08
**Probably missing citations and some questions**

I appreciate the author's idea of using score on the trajectory as regularizer. But from my view, is the main theorem of this paper, Theorem 2 (Instantaneous Change of Score Function) the same as Theorem D.1 in ScoreODE [1]? I hope the author can add some citations and discuss their relations.

Another thing is from the practical perspective: I think one of the main problems of FFJORD is time consumption, since solving the ode (Instantaneous Change of Log-likelihood) is highly intractable on high-dimensional case. The author's regularizer further needs to solve a more complex ode (Instantaneous Change of Score) when training, so I'm curious about the tractability on real datasets such as CIFAR-10. In this sense, I think the author's method doesn't solve the real critical problem of FFJORD. I hope the author can make some clarification on this question.

As far as I know, the recent training of CNF is mostly through score-based models, such as ScoreFlow [2] and ScoreODE [1]. I think it will be more convincing if the author can make more related comparisons, both theoretically and experimentally.

In summary, the author proposes a regularizer to stabilize the training of CNFs. While experimentally making improvements, I have some questions about its relationship with some recent works, especially score-based models, and the experimental advantage over them. I hope more demonstration about these questions can make this work more complete.

[1] Lu, Cheng, et al. "Maximum likelihood training for score-based diffusion odes by high order denoising score matching." International Conference on Machine Learning. PMLR, 2022.
[2] Song, Yang, et al. "Maximum likelihood training of score-based diffusion models." Advances in Neural Information Processing Systems 34 (2021): 1415-1428.

---

> ### Author Response · Authors · 2022-11-09
> **Thanks for your insightful comments and we would like to address your concerns and questions.**
>
> We thank the reviewer for providing valuable feedback to improve the work. We will address concerns and questions from the reviewer, and answer them individually.
>
> 1. The main purpose of proposed ACNF and ascent regularization is not to stabilize the training of CNFs, but to study some overlooked issues in the behavior of learned normalizing flows as presented in Figure 1 and Figure 2: 1) Vanilla CNFs are generally slow to converge to the target distribution if flow length for training is too lengthy, and thus have higher computation cost with respect to the gain of density estimation; 2) learned vanilla CNFs are sensitive to the length of flows and they diverge fast if flow length is larger than the training flow length; 3) hyper-parameter selection of $T$ has to be performed for deploying models in practice.
>
> 2. By presenting the definition of ACNF and the steepest dynamics in Theorem 1, we can show the motivation for maximum likelihood learning using a high-order score matching objective [1] beyond providing a better density estimation but also regularizing flows for better behavior, e.g. accelerating the density convergence, which can be critical for the online inference system.
>
> 3. We are fully aware that the naive implementation of maximum likelihood learning in Algorithm 1 requires higher computational cost compared to the baseline methods as mentioned in the scope and limitation section. More efficient implementations, e.g. lower computational cost estimator or specially designed NN architecture for Jacobian and gradients of divergence, are needed for scaling up for high-dimensional problems. [1] proposes another excellent alternative for achieving the same goal. We would like to experiment with it in the future.
>
> 4. In Section A.6 in Appendix of the updated manuscript, we reinterpret the ascent regularization as a score matching objective to build a connection to score-based models. Therefore, it is possible to implement Algorithm 1 and Algorithm 2 in more time efficient ways for training like [1,2].
>
> 5. We also like to point out one potential limitation of some score-based models: as the explicit score matching objective can hardly be used directly for training, most score-based models rely on its surrogates, e.g. denoising score matching. To enable the training, the objective relies on the explicit form of conditional (noised and denoised) distributions like Gaussian. For image or data generation tasks, the Gaussian assumption is not so constrained as long as the chain of discrete transformation is long enough for the adequate expressivity of the marginal distribution $\tilde{p}_T$. However, for inference tasks, e.g. use flows as variational approximation, the distributions induced by the flows with Gaussian assumption can hinder their approximate potential for true posterior. CNFs instead do not limit by the explicit form, which makes them more suitable for inference purposes.
>
> 6. We understand theorem 2 essentially reveals the same result as concurrent Theorem D.1 in [1] but using different techniques and we will cite [1] and clarify this in the updated version of the manuscript. However, to our knowledge, scoreODE does not utilize this theorem in the practical implementation while we show the integral of score function can be useful for ascent regularization.
>
> [1] Lu, Cheng, et al. "Maximum likelihood training for score-based diffusion odes by high order denoising score matching." International Conference on Machine Learning. PMLR, 2022.
> [2] Song, Yang, et al. "Maximum likelihood training of score-based diffusion models." Advances in Neural Information Processing Systems 34 (2021): 1415-1428.

---

> > ### Public Comment · ~Kaiwen_Zheng2 · 2022-11-10
> > **Thanks for your prompt and convincing reply**
> >
> > Thank the author for the detailed and sincere explanation for my public concerns. In the reply, the comparison with score-based models is very thorough, and now I gain a better understanding of their respective pros and cons. While theoretically there is trade-off between tractability and limited conditional distribution form, I hope one day the experimental demonstration about this phenomenon, and techniques to reduce the regularizer's computation overhead can be developed. I'm looking forward to the full version of the paper.

---

### Decision · Program_Chairs · 2023-01-20

**Decision:**

Accept: poster

**Justification For Why Not Higher Score:**

The experiments are limited to simple datasets, hence it remains unclear to what extent the advertised benefits translate to more complex data.

**Justification For Why Not Lower Score:**

The proposed method is of interest to the community of normalizing flows, it proposes a sound and novel method for improving training of continuous normalizing flows.

**Metareview: Summary, Strengths And Weaknesses:**

The paper proposes a way to regularize continuous normalizing flows by ensuring they have monotonically decreasing KL divergence wrt to some base distribution along the integration path. The paper empirically shows the proposed method leads to flows that converge faster and are more stable on tabular datasets and MNIST images

Overall the reviewers and the AC agree that the paper would be interesting to the community.

**Note From Pc:**

if the above contains the word "oral" or "spotlight" please see: "oral" presentation means -> notable-top-5% and "spotlight" means -> notable-top-25%. As stated in our emails, we are disassociating presentation type from AC recommendations

**Summary Of Ac-Reviewer Meeting:**

Through discussion between reviewers and authors on the platform, the reviewers reached a consensus of accepting the paper.